



# An Ensemble-Variational Inversion System for the Estimation of Ammonia Emissions using CrIS Satellite Ammonia Retrievals

Michael Sitwell[1] and Mark Shephard[1]

[1]Air Quality Research Division, Environment and Climate Change Canada, Toronto, Ontario, Canada

**Correspondence:** Michael Sitwell (michael.sitwell@ec.gc.ca)

**Abstract.** An ensemble-variational inversion system is developed for the estimation of ammonia emissions using ammonia retrievals from the Cross-track Infrared Sounder (CrIS) for use in the Global Environmental Multiscale - Modelling Air quality and Chemistry (GEM-MACH) chemical weather model. A novel hybrid method to compare logarithmic retrieval parameters to model profiles is presented. Inversions for the monthly mean ammonia emissions over North America were performed for May to August 2016. Inversions using the hybrid comparison method increased ammonia emissions at most locations within the model domain, with total monthly mean emissions increasing by 11–41%. The use of these revised emissions in GEM-MACH reduced biases with surface ammonia observations by as much as 25%. The revised ammonia emissions also improved the forecasts of total (fine+coarse) ammonium and nitrate and ammonium wet deposition, with biases decreasing by as much as 13%, but did not improve the forecasts of just the fine components of ammonium and nitrate. An additional area of $1.3 \times 10^5$ km$^2$ of upland forests in Canada were estimated to exceed the ecosystem's critical load due to the changes in ammonia deposition from the inversion. A comparison of biases resulting from inversions using different comparison methods shows favourable results for the hybrid comparison method.

## 1 Introduction

Ammonia (NH$_3$) is one of the most abundant reactive nitrogen species in the atmosphere. Major sources of ammonia include emissions from fertilizers, livestock, and biomass burning (Behera et al., 2013). Excess deposition of ammonia has been associated with eutrophication and soil acidification (Fangmeier et al., 1994; Krupa, 2003). As the primary basic gas in the atmosphere, ammonia plays a central role in the neutralization of acids and the formation of particulate matter (PM) in the atmosphere (Tsimpidi et al., 2007; Makar et al., 2009). And while ammonia has a relatively short atmospheric lifetime, on the order of hours to days (Van Damme et al., 2018), fine particulate matter can last much longer in the atmosphere, on the order of days to weeks (Seinfeld and Pandis, 2006). Particulate matter with diameters smaller than 2.5 $\mu$m (PM$_{2.5}$) has been associated with cardiovascular and respiratory disease and premature mortality (Pope III et al., 2002; Burnett et al., 2014; Lelieveld et al., 2015). Additionally, inorganic aerosols can affect climate directly through radiative forcing (Adams et al., 2001; Martin et al., 2004) and indirectly through its effect on cloud formation (Abbatt et al., 2006). In contrast to trends of nitrogen oxides (NO$_x$) and sulfur dioxide (SO$_2$) that have declined over the last few decades in many regions, ammonia levels


have either been constant or increasing in many parts of the world (Butler et al., 2016; Yao and Zhang, 2016; Warner et al., 2017).

Until recently, atmospheric ammonia observations were mainly limited to networks of ground-based measurements. The relatively short atmospheric lifetime of ammonia and the inhomogeneity of its sources results in a largely inhomogeneous distribution within the atmosphere. Accordingly, the limited spatial and temporal coverage of observations from these ground-based

networks limits their ability to constrain ammonia emissions. While bottom-up emissions inventories can provide emissions estimates at a high spatial resolution, they generally rely on a variety of land-use specifications (such as farming practices) and emission factors that can be highly uncertain at the spatial and temporal scales required for modern chemical transport models. Due to these factors, large uncertainties in the spatial and temporal distribution of ammonia emissions exist in current inventories (Clarisse et al., 2009; Heald et al., 2012; Walker et al., 2012).

More recently, the development of retrieval algorithms for atmospheric ammonia used with satellite-borne instruments such as the Tropospheric Emission Spectrometer (TES) (Shephard et al., 2011, 2015), the Infrared Atmospheric Sounding Interferometer (IASI) (Clarisse et al., 2009; Van Damme et al., 2014), the Atmospheric Infrared Sounder (AIRS) (Warner et al., 2016), and the Cross-track Infrared Sounder (CrIS) (Shephard and Cady-Pereira, 2015; Shephard et al., 2020) has led to a wealth of new information on atmospheric ammonia. The wide spatial and temporal coverage of these observations has greatly

improved our understanding of ammonia's global distribution (Clarisse et al., 2009; Shephard et al., 2011; Van Damme et al., 2014; Warner et al., 2016), seasonal variation (Shephard et al., 2011; Van Damme et al., 2014; Warner et al., 2016), long-term trends (Warner et al., 2017), presence in fire plumes (Coheur et al., 2009; Adams et al., 2019), large emissions point sources (Van Damme et al., 2018; Dammers et al., 2019), and dry deposition fluxes (Kharol et al., 2018).

The information contained within observational data can be blended with bottom-up emission inventories using inversion

methods. Previous works, such as Gilliland et al. (2003, 2006) and Paulot et al. (2014), used inversion techniques to obtain ammonia emission estimates using ammonium precipitation observations, although these precipitation-chemistry observation networks suffer from the same lack of spatial and temporal coverage as described above, as well as the added complication of only indirectly measuring ammonia levels through its wet deposition. However, the wide spatial and temporal coverage of ammonia observations made by satellite-borne instruments have the potential to greatly improve these estimates. Recently,

a number of studies have performed inversions for ammonia emissions via 4D-Var using the GEOS-Chem global chemistry transport model, such as Zhu et al. (2013) and Zhang et al. (2018) that used TES ammonia retrievals over the US and China, respectively, Cao et al. (2020) that used CrIS ammonia retrievals, and Li et al. (2019) that used simulated CrIS observations.

In this work, we develop an ammonia emissions inversion system using CrIS ammonia retrievals and Environment and Climate Change Canada's (ECCC) Global Environmental Multiscale - Modelling Air quality and Chemistry (GEM-MACH)

chemical weather model. While 4D-Var has been shown to be a useful technique for performing emissions inversions, it requires a model adjoint, which can be restrictive. The additional burden of maintaining the code for model adjoints has sparked interest in assimilation techniques that do not require the use of a model adjoint. In particular, due to the adoption at ECCC of ensemble methods that allow for flow-dependent background error covariances without the use of a model adjoint, the model adjoint that was previously used with 4D-Var at ECCC has been deprecated and is no longer maintained. As such, our inversion system





employs an ensemble-variational technique that has been successfully used in meteorological assimilation (Buehner et al., 2013, 2015; Caron et al., 2015) that does not require a model adjoint. The aim of this work is to (1) develop an ensemble-variational inversion system that is capable of refining the ammonia emissions currently used in GEM-MACH by using CrIS ammonia retrievals and (2) determine the impact of these updated emissions on the ammonia fields predicted by GEM-MACH as well as on fields related to inorganic PM.

This paper is organized as follows: Section 2 describes the CrIS ammonia retrievals utilized in the inversions, the surface observation networks used for validation, as well as the GEM-MACH model. The ensemble-variational method used for the emissions inversions, the ensemble used in the inversions, and possible methods for retrieval-to-model comparison are detailed in Section 3. Results are shown in Section 4 followed by conclusions in Section 5.

## 2   Observational Data and Air Quality Model

In this section, we describe the observation data sets and air quality model used in this study. We begin with a description of the CrIS ammonia retrievals used in the emissions inversions, followed by a brief discussion of surface observations used to validate the inversion results. This section concludes with a summary of the GEM-MACH model.

### 2.1   CrIS Ammonia Retrievals

CrIS is a Fourier transform spectrometer that is currently aboard the Suomi National Polar-orbiting Partnership (S-NPP; 
launched October 2011) and NOAA-20 (launched November 2017) satellites. As the year 2016 is the focus of this work, all observations used were from the S-NPP satellite. The S-NPP satellite, which is in a sun-synchronous orbit, has local over-pass times of about 01:30 and 13:30. CrIS scans across the satellite track with a swath width of ∼2200 km with a resolution of ∼14 km at nadir. CrIS measures the infrared spectrum over three different wavelength bands. As 960–970 $\mathrm{cm^{-1}}$ is the primary absorbing infrared spectral region for ammonia, the ammonia retrievals concentrate on the 650–1095 $\mathrm{cm^{-1}}$ (9.14–15.38 $\mu$m) 
band, where CrIS has a spectral resolution of 0.625 $\mathrm{cm^{-1}}$.

The CrIS Fast Physical Retrieval (CFPR) ammonia retrieval strategy (Shephard and Cady-Pereira, 2015), which is very similar to that used for TES ammonia retrievals (Shephard et al., 2011), uses an optimal estimation method that minimizes the difference between observed spectral radiances and those generated by a nonlinear radiative transfer model, with an added a priori regularization term. The ammonia profiles are retrieved on 14 pressure levels. An a priori profile is selected from a set 
of three profiles that aim to represent typical profiles from polluted, moderately polluted, and unpolluted areas.

Figure 1 shows the retrieval of an ammonia profile and its associated averaging kernels to illustrate a typical retrieval. From the averaging kernels displayed in the right panel of Fig. 1, we can see that the sensitivity of this particular retrieval peaks around 900 hPa. This is typical for the CrIS ammonia retrievals, which generally have a peak sensitivity below 700 hPa (Shephard et al., 2020).



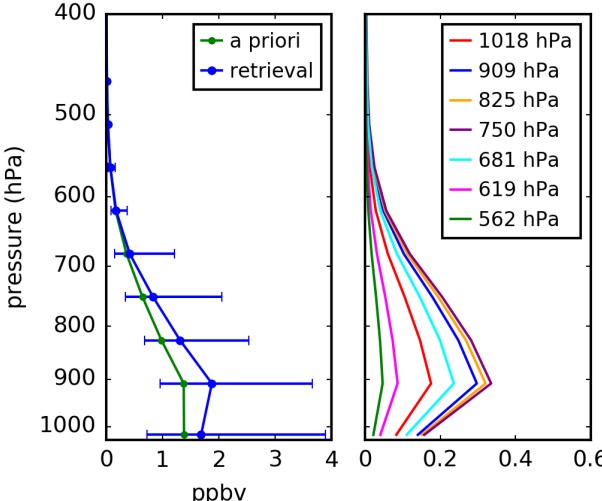

**Figure 1.** CrIS ammonia retrieval at (42.21°N, 70.82°W) on 12 May 2016 17:31:59 UTC. The left panel shows the retrieval and a priori profiles, while the right panel shows the averaging kernels for the seven vertical levels closest to the surface. The number of degrees of freedom for this retrieval is 0.956.

For the emissions inversions, we utilized CrIS retrievals within the GEM-MACH regional domain, which consists mostly of North America. Additionally, we only consider retrievals over land. As a quality control measure, we only use CrIS ammonia retrievals that have a signal-to-noise ratio of at least one and have at least 0.1 degrees of freedom.

    The inversions in this work use retrievals between May and August 2016, when the ammonia signal is relatively strong, as to maximize the number of CrIS retrievals that have an adequately large signal-to-noise ratio. The particular months chosen
also reflect the availability of required input files for GEM-MACH. The year 2016 was chosen in part due to the relatively low number of forest fires during this year (Munoz-Alpizar et al., 2017; Earl and Simmonds, 2018) to decrease the chances of misattributing elevated ammonia levels due to forest fires with other emission sources.

    During the course of this work, an issue related to accounting for the non-detection of ammonia in the CrIS spectrum was identified, which primarily affected the CrIS data in non-source regions (e.g. high northern latitudes). As these locations were
not areas of interest for this work, only observations south of 60°N were used in this study. While non-detects have been accounted for in newer versions of the CrIS ammonia product (White et al., 2021), these corrections were only made available after the completion of this study.

    Although CrIS provides twice-daily ammonia observations, the uncertainties on individual retrievals are relatively large (Shephard and Cady-Pereira, 2015). Thus, while inversions on the time scale of a day could be performed, increasing the
number of observations used in a single inversion by increasing the time window of the inversion can mitigate the large uncertainties on individual retrievals. Accordingly, each inversion used a whole month's worth of CrIS retrievals and yielded an estimate of the monthly mean ammonia emissions. The total number of retrievals used for each inversion and their spatial



distribution are shown in Figure S1 of the Supplement. The selection process described in the previous paragraphs resulted in 70–85% of the retrievals used in the inversions coming from daytime retrievals.

## 2.2 Surface Air Quality Observations

The GEM-MACH surface fields predicted using the original emissions and those from the inversion were compared to surface observations from various observation networks. In addition to ammonia observations, to examine the impact of the inversion-based emission changes on inorganic PM formation, GEM-MACH was also compared to observations of nitric acid ($HNO_3$), sulfur dioxide, ammonium ($NH_4^+$), nitrate ($NO_3^-$), and sulfate ($SO_4^{2-}$). For the aerosol species, we distinguish observations of the total concentration over all particle sizes (p-$NH_4$, p-$NO_3$, and p-$SO_4$ for ammonium, nitrate, and sulfate, respectively) from observations only measuring the $PM_{2.5}$ component ($PM_{2.5}$-$NH_4$, $PM_{2.5}$-$NO_3$, and $PM_{2.5}$-$SO_4$ for ammonium, nitrate, and sulfate, respectively). A summary of these observations is shown in Table 1 and site locations are shown in Figure S2 of the Supplement.

The Ammonia Monitoring Network (AMoN; http://nadp.slh.wisc.edu/amon) is one of several monitoring networks that is part of the US National Atmospheric Deposition Program (NADP), with most sites located in the US. AMoN monitors ammonia levels using Radiello®passive diffusion samplers that are deployed for two-week periods. Canada's National Air Pollution Surveillance (NAPS; https://www.canada.ca/en/environment-climate-change/services/air-pollution/monitoring-networks-data/national-air-pollution-program.html) network provides observations of a variety of pollutants, including ammonia, within Canada. NAPS uses a citric-acid-coated denuder to collect ammonia samples over a 24-hour period every three or six days.

Both the US Environmental Protection Agency's (EPA) Clean Air Status and Trends Network (CASTNET; https://www.epa.gov/castnet) and ECCC's Canadian Air and Precipitation Monitoring Network (CAPMoN; https://www.canada.ca/en/environment-climate-change/services/air-pollution/monitoring-networks-data/canadian-air-precipitation.html) use 3-stage filter packs to collect $HNO_3$, $SO_2$, p-$NH_4$, p-$NO_3$, and p-$SO_4$. There are currently two major networks in the US that monitor speciated $PM_{2.5}$ levels: The US EPA's Chemical Speciation Network (CSN; https://www.epa.gov/amtic/chemical-speciation-network-csn) and the Interagency Monitoring of PROtected Visual Environments (IMPROVE; http://vista.cira.colostate.edu/Improve) network (partnership between with the US EPA, US National Park Service, and other federal, state, and tribal agencies). CSN has a mission focused on air quality and has stations primarily located in urban settings, while IMPROVE concentrates on visibility and has stations mainly in rural locations. CSN and IMPROVE both use nylon filters to collect $PM_{2.5}$-$NO_3$ and $PM_{2.5}$-$SO_4$ (and $PM_{2.5}$-$NH_4$ in the case of CSN). Speciated $PM_{2.5}$ observations are also measured at NAPS sites, with $PM_{2.5}$-$NH_4$, $PM_{2.5}$-$NO_3$ and $PM_{2.5}$-$SO_4$ collected by Teflon filters. NAPS also provides hourly observations of total $PM_{2.5}$. Additional hourly total $PM_{2.5}$ measurements are supplied by the EPA's Air Quality System (AQS; https://www.epa.gov/airdata), which collects data from EPA and state, local, and tribal agencies.

Precipitation-chemistry concentration measurements for ammonium, nitrate, and sulfate were also examined. Observations from two NADP networks were considered: The National Trends Network (NTN; http://nadp.slh.wisc.edu/ntn) and the Atmospheric Integrated Research Monitoring Network (AIRMoN; http://nadp.slh.wisc.edu/airmon). Although the AIRMoN pro-



| network | species | medium | collection substrate/ apparatus | analytical method | sampling period | sampling frequency | # of stations |
|---|---|---|---|---|---|---|---|
| NAPS | $NH_3$ | air | CACD | IC | 24 h | 3 or 6 days | 12 |
| | $PM_{2.5}$-$NH_4$, $PM_{2.5}$-$SO_4$ | air | TF | IC | 24 h | 3 or 6 days | 26 |
| | $PM_{2.5}$-$NO_3$ | air | TF, NF | IC | 24 h | 3 or 6 days | 12 |
| | $PM_{2.5}$ | air | GFFT | BAM | 1 h | 1 h | 205 |
| | | | TCGF | TEOM | 1 h | 1 h | 9 |
| AMoN | $NH_3$ | air | PDS | color | 2 weeks | 2 weeks | 102 |
| CSN | $PM_{2.5}$-$NH_4$, $PM_{2.5}$-$SO_4$ | air | NF | IC | 24 h | 3 or 6 days | 140 |
| | $PM_{2.5}$-$NO_3$ | air | NF | IC | 24 h | 3 or 6 days | 139 |
| IMPROVE | $PM_{2.5}$-$NO_3$, $PM_{2.5}$-$SO_4$ | air | NF | IC | 24 h | 3 or 6 days | 154 |
| AQS | $PM_{2.5}$ | air | GFFT | BAM | 1 h | 1 h | 346 |
| | | | TCGF | TEOM | 1 h | 1 h | 56 |
| CASTNET | $HNO_3$ | air | NF | IC | 1 week | 1 week | 95 |
| | $SO_2$ | air | NF, CF | IC | 1 week | 1 week | 95 |
| | p-$NO_3$, p-$SO_4$ | air | TF | IC | 1 week | 1 week | 95 |
| | p-$NH_4$ | air | TF | color | 1 week | 1 week | 95 |
| CAPMoN | $HNO_3$ | air | NF | IC | 24 h | 24 h | 20 |
| | $SO_2$ | air | CF | IC | 24 h | 24 h | 20 |
| | p-$NH_4$, p-$NO_3$, p-$SO_4$ | air | TF | IC | 24 h | 24 h | 20 |
| | $NO_3$, $SO_4$ | precip | PC | IC | 24 h | 24 h | 32 |
| | $NH_4$ | precip | PC | color | 24 h | 24 h | 32 |
| NTN | $NO_3$, $SO_4$ | precip | PC | IC | 1 week | 1 week | 261 |
| | $NH_4$ | precip | PC | color | 1 week | 1 week | 261 |
| AIRMoN | $NO_3$, $SO_4$ | precip | PC | IC | 24 h | - | 6 |
| | $NH_4$ | precip | PC | color | 24 h | - | 6 |

**Table 1.** Surface observations used in this study. Collection substrate/apparatus types include Teflon filter (TF), nylon filter (NF), $K_2CO_3$ impregnated cellulose filter (CF), Teflon-coated glass fiber filter (TCGF), glass fiber filter tape (GFFT), citric-acid-coated-denuder (CACD), passive diffusion sampler (PDS), and precipitation collectors (PC). Analytical methods include ion chromatography (IC), colorimetry (color), beta attenuation monitoring (BAM), and tapered element oscillating microbalance (TEOM). AIRMoN precipitation samples are collected within 24 hours of the start of a precipitation event.

gram ceased operations in 2019, observations from this network were available for 2016, the year examined in this work. Precipitation-chemistry measurements in Canada are provided by the CAPMoN network.



To evaluate the predicted fields from GEM-MACH against these surface observations, we examine the changes in the normalized mean bias (NMB), normalized standard deviation of differences (NSTD), and the Pearson correlation coefficient ($\rho$)

between GEM-MACH and the observations, defined by

$$\text{NMB} = \frac{100\%}{\bar{O}} \times \frac{1}{N} \sum_i^N (M_i - O_i), \tag{1a}$$

$$\text{NSTD} = \frac{100\%}{\bar{O}} \times \sqrt{\frac{1}{N-1} \sum_i^N (M_i - O_i - \bar{M} + \bar{O})^2}, \tag{1b}$$

$$\rho = \frac{\sum_i^N (M_i - \bar{M})(O_i - \bar{O})}{\sqrt{\sum_i^N (M_i - \bar{M})^2}\sqrt{\sum_i^N (O_i - \bar{O})^2}}, \tag{1c}$$

$$\bar{O} = \frac{1}{N} \sum_i^N O_i, \tag{1d}$$

$$\bar{M} = \frac{1}{N} \sum_i^N M_i, \tag{1e}$$

where $O_i$ and $M_i$ are the $i^{\text{th}}$ observation and its corresponding model value, respectively, and $N$ is the number of observations. Some observations from networks that have a sampling period of a week or longer (AMoN, CASTNET, NTN) straddled two months, i.e. started the end of one month and finished at the beginning of the next month. For monthly statistics of concentration observations, if an observation had at least four days within a month it is included in the statistics calculation for that month. For precipitation-chemistry concentration observations, observations straddling months are not included in the monthly statistics.

**2.3 The GEM-MACH Model**

GEM-MACH (Moran et al., 2010; Gong et al., 2015; Pavlovic et al., 2016; Pendlebury et al., 2018), ECCC's operational air quality model, is an online chemical transport model embedded within the Global Environmental Multiscale (GEM) model (Côté et al., 1998b, a; Girard et al., 2014), ECCC's operational weather forecasting model. GEM-MACH augments the GEM model by adding tropospheric gas-phase, aqueous-phase, and inorganic heterogeneous chemistry to the model.

Both regional and global versions of GEM-MACH are available. We use version 2.4.6-LTS.17 of the regional GEM-MACH model (Moran et al., 2018, 2019), which is based on version 4.8-LTS.17 of GEM (Girard et al., 2014). The model domain encompasses Canada, the United States, and northern Mexico. The horizontal grid has a spacing of $0.09°$ ($\sim$10 km), while the vertical grid has 80 hybrid log hydrostatic pressure levels on a Charney-Phillips staggered grid (Charney and Phillips, 1953; Girard et al., 2014) that extends from the surface to 0.1 hPa. In this work, GEM-MACH is run with sequential 12-hour





forecasts, where the meteorological fields are refreshed at the beginning of each forecast using analyses from ECCC's Regional
Deterministic Prediction System (RDPS) (Caron et al., 2015).

The gas-phase chemistry in GEM-MACH is modeled using the ADOM-II mechanism (Stockwell and Lurmann, 1989; Stroud
et al., 2008). The gas-phase dry deposition scheme is based on the unidirectional 'big leaf' resistance model of Zhang et al.
(2002). While this unidirectional model is currently used in the operational version of GEM-MACH, recently a bidirectional

flux model for ammonia has been added to GEM-MACH (Whaley et al., 2018). The use of this bidirectional flux scheme
instead of the unidirectional model will be the subject of future work. Particle dry deposition and gravitational settling is
modeled using the scheme of Zhang et al. (2001) and the wet deposition scheme is described in Gong et al. (2006).

The aerosol chemical components considered by GEM-MACH are sulfate, nitrate, ammonium, elemental carbon, crustal
material, sea salt, primary and secondary organic aerosols, and $H_2O$. GEM-MACH uses a sectional representation of the PM

size distribution and currently can be run with either 2 or 12 size bins. While the 2-bin model only gives limited information
about the aerosol size distribution (one 0–2.5 $\mu$m fine size bin and one 2.5–10 $\mu$m coarse-fraction bin), it offers increased
computational speed and reduced memory requirements as compared to the 12-bin model. Currently, the operational version
of GEM-MACH runs the 2-bin version of the model. As this work will in part be used to evaluate the potential use of CrIS
ammonia retrievals in an operational air-quality model, the 2-bin model was used for this work.

Inorganic heterogeneous chemistry is implemented in GEM-MACH using the HETV code (Makar et al., 2003), based
on the ISORROPIA model (Nenes et al., 1998), which computes the gas/aerosol phase partitioning for the sulfate-nitrate-
ammonium system. HETV computes this partitioning using the assumption of equilibrium to determine the abundances of
$NH_3(g)$, $HNO_3(g)$, $NO_3^-(aq)$, $SO_4^{2-}(aq)$, $NH_4^+(aq)$, $HSO_4^-(aq)$, $NH_4NO_3(s)$, $NH_4HSO_4(s)$, $(NH_4)_3H(SO_4)_2(s)$, and
$(NH_4)_2SO_4(s)$. While the HETV code computes the partitioning for all of the aforementioned species, for the aqueous- and

solid-phase species, only the total mass of sulfate, nitrate, and ammonium within each size bin is retained in GEM-MACH after
HETV is run. To offset the relatively large computational expense of the heterogeneous chemistry done by HETV, HETV com-
putes the gas/aerosol partitioning using a single 'bulk' calculation using the total mass of each species over all aerosol bin sizes.
Following this bulk calculation, the change in bulk mass is distributed to the different size bins using ratios of gas-to-particle
diffusion rates, computed via the formulation of Fuchs and Sutugin (1970) (see also Makar et al. (1998)).

Emissions are supplied to GEM-MACH via hourly, gridded, speciated emissions files. These emissions files were created
using the Sparse Matrix Operator Kernel Emissions (SMOKE; http://www.smoke-model.org/index.cfm) processing system,
which assigns temporal profiles, spatial surrogate fields, and speciation profiles based on source type to monthly or annual an-
thropogenic emissions inventories (Moran et al., 2018). The base case model run in this study used the set of emissions used by
the operational version of GEM-MACH at the time of writing (version 3.1.2 of this emissions data set). These emissions were

constructed using a 2013 Canadian emissions inventory from Canada's Air Pollutant Emissions Inventory (https://www.canada.
ca/en/environment-climate-change/services/pollutants/air-emissions-inventory-overview.html), a 2017 US projected inventory
obtained from the EPA's Air Emissions Modeling 2011 Platform (version 6.3; https://www.epa.gov/air-emissions-modeling/
2011-version-63-platform), and a 2008 Mexican inventory also obtained from the EPA's Air Emissions Modeling 2011 Plat-





form (version 6.2; https://www.epa.gov/air-emissions-modeling/2011-version-62-platform). While not done in this study,
these inventory-based emissions can be supplemented by other emissions types, such as forest fire emissions.

## 3  Emissions Inversion Procedure

This section details different aspects of the inversion process. First, the ensemble-variational algorithm used to perform the
emissions inversions is described. This is followed by a description of the ensemble used in the inversions. Lastly, a novel
method for comparing model profiles to satellite retrievals in the inversion algorithm is developed.

### 3.1  The Ensemble-Variational Method

Variational methods solve inverse problems through the minimization of a cost function to obtain an updated solution, known
as the *analysis*, that is a statistical blend of observational and background (a priori or 'first-guess') information. The relative
influence the observational and background information have on the analysis is dictated by the specification of error covariances
for the observations and background. Ensemble-variational techniques, commonly referred to as *EnVar*, use an ensemble of
model states to model the background error covariance. In this section (and Appendix A), we review the EnVar method and
apply it an emissions inversion.

Most variational methods solve for the additive field that yields the analysis, referred to as the *increment*, instead of the
analysis itself. In our case, we seek the atmospheric ammonia concentration increment vector $\Delta \mathbf{c}$ that minimizes the cost
function $J$ defined by

$$J = \frac{1}{2}\Delta\mathbf{c}^{\mathrm{T}}\mathbf{B}_{cc}^{-1}\Delta\mathbf{c} + \frac{1}{2}(\mathbf{d} - \mathbf{H}\Delta\mathbf{c})^{\mathrm{T}}\mathbf{R}^{-1}(\mathbf{d} - \mathbf{H}\Delta\mathbf{c}), \tag{2}$$

where $\mathbf{B}_{cc}$ is the atmospheric concentration background error covariance matrix and $\mathbf{R}$ is the observation error covariance
matrix. The vector $\mathbf{d} = \mathbf{y} - H(\mathbf{c}^{\mathrm{b}})$, known as the *innovation*, is the difference between the observations $\mathbf{y}$ and the background
concentration $\mathbf{c}^{\mathrm{b}}$ input into the observation operator $H$. The observation operator $H$, and its linearization $\mathbf{H}$ that appears in Eq.
(2), relates the atmospheric concentrations to the observations. For the cost function given in Eq. (2), the first term measures
the mismatch between the model state and the background, while the second term measures the mismatch between the model
state and the observations.

The atmospheric concentrations produced by a model are dependent on the emissions as well as many other factors such
as meteorology, initial conditions, and boundary values. For simplicity, in this work we assume that the uncertainty in the
atmospheric concentrations is due solely to the uncertainty in emissions and neglect all other sources of uncertainty, such as
that from meteorology and chemistry. We also assume that our observations are of the atmospheric chemical concentrations
only and do not directly measure emissions. In this situation, we can use the properties of *analysis splitting* to relate the
increment in the unobserved emissions $\Delta \mathbf{e}$ to the increment in the observed concentrations $\Delta \mathbf{c}$ by (Ménard et al., 2019)

$$\Delta\mathbf{e} = \mathbf{B}_{ec}\mathbf{B}_{cc}^{-1}\Delta\mathbf{c}, \tag{3}$$





where $\mathbf{B}_{ec}$ is the error cross-covariance between the emissions and the atmospheric concentrations. In this work, we define $\mathbf{e}$

and its increment $\Delta\mathbf{e}$ to allow them to represent either the emissions directly or a parameterization of the emissions, which will be elaborated on in the next section. The details of the minimization of the cost function in Eq. (2) and the subsequent transformation to the emissions increment in Eq. (3) are given in Appendix A.

Ensemble-variational methods use an ensemble of states to model the background error covariances required in Eq. (2). The background error covariances can originate solely from an ensemble, as done in this work, or can be a hybrid that combines

ensemble information with information from other sources. For this work, we use an ensemble of 100 members, with each ensemble member $i$ consisting of a perturbed emissions field $\delta\mathbf{e}_i$ and a corresponding perturbed concentration field $\delta\mathbf{c}_i$ generated by running GEM-MACH with the perturbed emissions. The way in which the emissions were perturbed will be described in the following section.

To minimize the effect of spurious long-distance correlations due to a small ensemble size, as well as to increase the rank

of the background error covariance matrix, the concentrations and emissions background error covariances are localized via the Hadamard (element-wise) product with localization matrices $\mathbf{L}_{cc}$ and $\mathbf{L}_{ee}$, respectively (see Appendix A for more details). For the concentrations, we use a localization $\mathbf{L}_{cc}$ that is separable in the horizontal and the vertical. We construct $\mathbf{L}_{ee}$ and the horizontal component of $\mathbf{L}_{cc}$ from homogeneous and isotropic functions with correlations modeled with the fifth-order function of Gaspari and Cohn (1999). We choose $\mathbf{L}_{cc}$ and $\mathbf{L}_{ee}$ to have horizontal correlations with half-widths at half-maximum of 120

km and 100 km, respectively, and $\mathbf{L}_{cc}$ to have vertical correlations with a half-width at half-maximum of 3 km. These values were chosen to be moderately larger than estimates of the distances traveled during the atmospheric lifetime of ammonia (Dammers et al., 2019).

### 3.2  Emissions Parameterization and Ensemble Construction

As mentioned in Section 2.1, the inversions performed in this study solve for monthly mean ammonia emissions, which were

specified on the same horizontal grid used for the GEM-MACH emissions ($\sim$10 km resolution). In emissions inversion studies, it is common to use a parameterization of the emissions, as it may be advantageous to solve for the parameterization variables instead of the emissions directly. Additionally, a parameterization of the emissions may be convenient for generating the ensemble of emissions required for the ensemble-variational inversion. The use of such a parameterization for both of the aforementioned purposes is discussed in this section.

In Section 3.1, we defined $\mathbf{e}$ flexibly to allow it to represent either the emissions directly or a parameterization of the emissions. Here we denote the actual emissions by $E$ to differentiate it from a (possible) parameterization variable $e$ and define $E^{(0)}$ and $E^{(1)}$ as the original and updated/perturbed emissions, respectively. If no parameterization is used then we have $E^{(1)} = E^{(0)} + \Delta e$. Often a scaling factor is used to parameterize the emissions, most often in a linear ($E^{(1)} = (1 + \Delta e)E^{(0)}$) or exponential ($E^{(1)} = \exp(\Delta e)E^{(0)}$) form. One benefit of the exponential scaling form is that the scaling factor is positive for

all values of $\Delta e$.

An initial comparison of the CrIS ammonia retrievals to GEM-MACH suggested that the processed ammonia emissions inventories described in Section 2.3 were missing emission sources in some locations. As the processed emissions inventory





at these locations may have a negligible value, if a scaling factor parameterization is used, an extremely large scaling factor value may have to be applied at these locations to achieve an adequately large posterior emissions value. As very large scaling values may cause numerical difficulties within the inversion computation, inversions that use scaling factor parameterizations often impose an upper bound on the scaling factor value, which in practice restricts the inversion to modifying only previously known emission sources.

To allow the inversion to add previously unknown emission sources, as well as restrict the emissions to positive values, we use an alternative emissions parameterization, given by

$$E^{(1)}(E^{(0)}, \Delta e) = \begin{cases} E^{(0)} + \Delta e & E^{(0)} + \Delta e \geq \varepsilon \\ \varepsilon \left( E^{(0)}/\varepsilon \right)^{(E^{(0)} + \Delta e - \varepsilon)/(E^{(0)} - \varepsilon)} & \text{otherwise} \end{cases}, \tag{4}$$

where the variable $\Delta e$ parameterizes the emissions and $\varepsilon$ is a positive constant. When $E^{(0)} + \Delta e$ is larger than $\varepsilon$, $\Delta e$ acts as a linear perturbation, which avoids the missing-source scaling problem described above. To avoid negative emissions values, if $E^{(0)} + \Delta e$ is smaller than $\varepsilon$, then the initial emissions $E^{(0)}$ are exponentially damped. The exponential dampening parameters were chosen so that the function $E^{(1)}(E^{(0)}, \Delta e)$ is continuous and has $E^{(1)}(E^{(0)}, \Delta e = 0) = E^{(0)}$ for all values of $E^{(0)}$. As our inversions solve for monthly mean ammonia emissions, $E^{(0)}$ represents the original monthly mean ammonia emissions (described in Section 2.3) and $E^{(1)}$ represents either the posterior or perturbed monthly emissions (depending on context). We choose a value of $0.1 \text{ g s}^{-1} \text{ cell}^{-1}$ ($\sim 0.26 \text{ tonnes month}^{-1} \text{ cell}^{-1}$) for the constant $\varepsilon$ (see Fig. 5 for comparison to monthly mean emissions).

Inversions based on ensemble-variational methods use an ensemble to approximate the full error covariance associated with our a priori knowledge of the model state. The ensemble of emissions were generated as follows: For each ensemble member $i$, a 2D random field $\delta e_i(\mathbf{x}_j)$ is generated at each grid point $j$ (with position $\mathbf{x}_j$). The random field $\delta e_i(\mathbf{x}_j)$, along with the original monthly mean ammonia emissions $E^{(0)}(\mathbf{x}_j)$, are used in Eq. (4) to yield a perturbed monthly mean ammonia emissions field $E_i^{(1)}(\mathbf{x}_j) = E^{(1)}(E^{(0)}(\mathbf{x}_j), \Delta e = \delta e_i(\mathbf{x}_j))$ at each grid point $j$ for each ensemble member. The ensemble of atmospheric chemical concentrations is then gained by generating hourly emissions for each member by applying the same temporal profiles as described in Section 2.3 to the perturbed monthly mean emissions and then running GEM-MACH with each set of hourly emissions.

The distribution from which the random fields $\delta \mathbf{e}$ are drawn determines the emissions background error covariance $\mathbf{B}_{ee}$. As is often the case in inversion/assimilation work, deriving background error covariances that accurately specify the uncertainties of our a priori knowledge of the emissions is difficult. For simplicity, to create our ensemble of emissions we draw $\delta \mathbf{e}$ from an isotropic normal distribution with a correlation half-width at half-maximum of 40 km. We set the standard deviations of this distribution to be 50% of the monthly mean values from the processed emissions inventory, but impose a minimum standard deviation value of $1 \text{ g s}^{-1} \text{ cell}^{-1}$ ($\sim 2.6 \text{ tonnes month}^{-1} \text{ cell}^{-1}$) so that areas without previously known emissions sources have a non-negligible ensemble variance.





### 3.3 Comparison Between Model Profiles and Satellite Retrievals

Variational inversion methods, such as the ensemble-variational method described in Section 3.1, rely on a comparison of a model with observations to generate an optimized state, done through the computation of the innovation $\mathbf{d} = \mathbf{y} - H(\mathbf{c}^{\mathrm{b}})$. For some observation data sets, such as many in situ observations, performing this comparison is trivial. But for some remotely-sensed observations, added complications can arise that make this comparison more difficult. In this section, we develop a new hybrid approach for comparing the CrIS ammonia retrievals to model profiles for use in an inversion system.

The CrIS ammonia retrievals are performed with the log of concentrations, which can be represented by (Rodgers, 2000)

$$\ln(\mathbf{c}^{\mathrm{r}}) = \mathbf{A}^{\log} \ln(\mathbf{c}) + (\mathbf{I} - \mathbf{A}^{\log}) \ln(\mathbf{c}^{\mathrm{a}}) + \mathbf{G}\epsilon, \tag{5}$$

where $\mathbf{c}$, $\mathbf{c}^{\mathrm{r}}$, and $\mathbf{c}^{\mathrm{a}}$ are the true, retrieved, and a priori volume mixing ratio profiles in ppmv, respectively, $\mathbf{A}^{\log}$ is the averaging kernel in log-space, $\mathbf{G}$ is the gain matrix (quantifies the sensitivity of the retrieval to the observation), and $\epsilon$ is the total error. By replacing the true atmospheric profile $\mathbf{c}$ with the model profile $\mathbf{c}^{\mathrm{b}}$ in Eq. (5), neglecting the noise term $\mathbf{G}\epsilon$, and taking the

exponent of both sides, one can construct an observation operator $H^{\log}$ that maps the model profile into a quantity that can be directly compared to the CrIS ammonia retrieval as

$$[H^{\log}(c^{\mathrm{b}})]_i = \prod_j (c_j^{\mathrm{b}})^{A_{i,j}^{\log}} \times (c_j^{\mathrm{a}})^{\delta_{i,j} - A_{i,j}^{\log}}, \tag{6}$$

where $i, j$ indexes the vertical levels of the profiles, $\delta_{i,j}$ is the Kronecker delta function, and $H^{\log}(\mathbf{c}^{\mathrm{b}})$ has units of ppmv. With this observation operator, the model equivalent to the retrieved profile is a product of profiles, raised to a power determined by

the averaging kernel.

While the observation operator in Eq. (6) functions well in many situations, problems may occur when part of the model profile $\mathbf{c}^{\mathrm{b}}$ is either zero or negligibly small. As the operator in Eq. (6) is a product of model profile terms, if the model profile is zero at any point in the profile, the whole product will be zero unless the corresponding component of the averaging kernel is exactly zero (or will approach an infinite value if the averaging kernel component is negative). This will occur regardless of

the values elsewhere in the model profile. An example of this behaviour can be seen in Figure 2, where the the model profile $\mathbf{c}^{\mathrm{b}}$ shown has non-negligible values below ~680 hPa, but are zero above ~600 hPa. Consequently, using Eq. (6) maps this model profile to the profile $H^{\log}(\mathbf{c}^{\mathrm{b}})$ which is zero at all vertical levels. As the $H^{\log}(\mathbf{c}^{\mathrm{b}})$ profile would be the same regardless of the model values below ~680 hPa, this observation operator may be problematic when comparing model profiles to retrievals since $H^{\log}(\mathbf{c}^{\mathrm{b}})$ in this case will be completely insensitive to the amount of ammonia in the model at lower levels.

To alleviate this issue, one might attempt to truncate these profiles to cut off all levels where the model profile is zero (or very small). However, it was observed that when GEM-MACH is used with the emissions inventory described in Section 2.3, in these situations the vertical levels at which the model profile is zero or very small often corresponded to levels where the averaging kernel is non-negligible and in some cases even peaks at these levels. Therefore, truncating the profiles in this manner would remove important vertical levels in the retrieval. Alternatively, one could retain all vertical levels but impose some model

profile minimum value, so that the terms in the product in Eq. (6) are multiplied by a small value instead of zero, yielding a

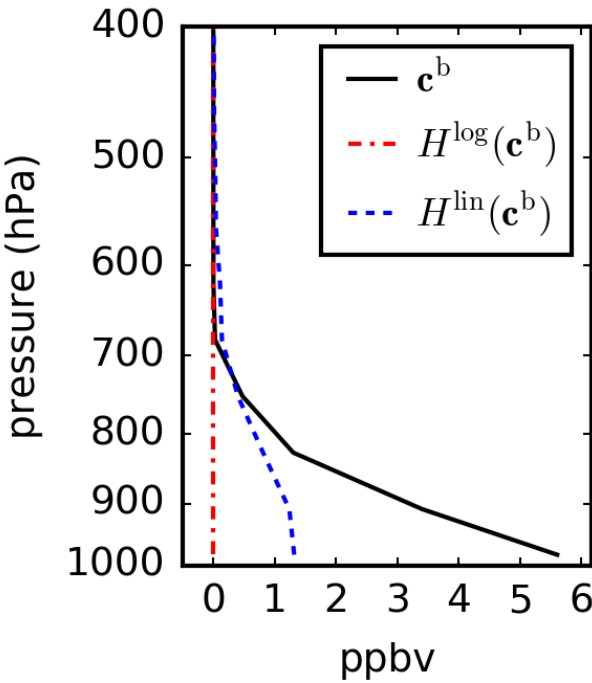

**Figure 2.** GEM-MACH profile at a retrieval location for the CrIS retrieval at (44.92°N, 92.96°W) on 12 May 2016 07:48:58 UTC. Profiles are shown for the model ($\mathbf{c}^{\mathrm{b}}$), as well as the profiles that results from applying the log-space ($H^{\log}$) and linear-space ($H^{\lin}$) observation operators.

non-zero value for $H^{\log}(\mathbf{c}^{\mathrm{b}})$. But in this case, $H^{\log}(\mathbf{c}^{\mathrm{b}})$ would primarily be determined by the somewhat arbitrarily chosen lower-bound value and would be relatively insensitive to the amount of ammonia in the model.

An observation operator that can avoid this problem is the linearization of Eq. (5), which can be written as

$$H^{\lin}(\mathbf{c}^{\mathrm{b}}) = \mathbf{A}^{\lin}\mathbf{c}^{\mathrm{b}} + (\mathbf{I} - \mathbf{A}^{\lin})\mathbf{c}^{\mathrm{a}}, \tag{7}$$

where $\mathbf{A}^{\lin}$ is the averaging kernel linearized in $\mathbf{c}^{\mathrm{b}} - \mathbf{c}^{\mathrm{a}}$. As $H^{\lin}(\mathbf{c}^{\mathrm{b}})$ is linear in $\mathbf{c}^{\mathrm{b}}$ instead of containing products of profiles, the zero-valued profiles that are produced with $H^{\log}(\mathbf{c}^{\mathrm{b}})$ are avoided. This is illustrated in Fig. 2, where $\mathbf{c}^{\mathrm{b}}$ is mapped to a non-zero profile $H^{\lin}(\mathbf{c}^{\mathrm{b}})$.

While using the linearized averaging kernel may avoid some of the problems encountered when using the log-space averaging kernel, the linearized averaging kernel may only be a good approximation when the difference between the model and a priori profiles is small. As only three a priori profiles are used in the CrIS ammonia retrieval, this will often not be the case.

One might consider only using the linearization when the difference between the model and a priori profiles is small, but this will likely be very restrictive and would exclude many profiles of interest, in particular cases where the model is significantly smaller than the selected a priori. On the other hand, using the linearization in all cases could introduce new biases into the





retrievals. For instance, when $H^{\mathrm{lin}}(\mathbf{c}^{\mathrm{b}})$ is compared to $H^{\log}(\mathbf{c}^{\mathrm{b}})$ for cases where $H^{\log}(\mathbf{c}^{\mathrm{b}})$ is not zero at all levels, $H^{\mathrm{lin}}(\mathbf{c}^{\mathrm{b}})$ is
systematically overestimated as compared to $H^{\log}(\mathbf{c}^{\mathrm{b}})$ and can introduce biases on the order of 0.5 ppbv.

To balance these different factors, we propose using a hybrid log/linear observation operator that attempts to choose the best method according to the particular situation. To assess which method to use, for each retrieval we compute the total column value of the model profile $X(\mathbf{c}^{\mathrm{b}})$ and the total column value of its log-space mapped profile $X(H^{\log}(\mathbf{c}^{\mathrm{b}}))$, where $X(\cdot)$ is the total column operator. We associate cases where the log-space observation operator has zeroed-out a non-negligible model
profile with model profiles that have a total column value $X(\mathbf{c}^{\mathrm{b}})$ above some specified minimum threshold value $X_{\min}$, as well as a maximum profile value above some minimum value $c_{\min}$, but that have a $X(H^{\log}(\mathbf{c}^{\mathrm{b}}))$ value that is below $X_{\min}$. If these criteria determine that the model profile has been zeroed-out, then the linearized observation operator is used, otherwise the log-space observation operator is used.

Most of the vertical distribution information in the CrIS ammonia retrieval originates from a priori information, so for
computational simplicity, we use the total column values computed from each profile for the comparison of the model to the retrieval in the inversion. However, we emphasize that the full averaging kernel over all vertical levels is nevertheless utilized. In summary, the CrIS retrieval is compared to the GEM-MACH model by computing the difference between the total column of the CrIS retrieval and the GEM-MACH model profile used in the piece-wise function

$$H(\mathbf{c}^{\mathrm{b}}) = \begin{cases} X(H^{\mathrm{lin}}(\mathbf{c}^{\mathrm{b}})) & X_{\min} < X(\mathbf{c}^{\mathrm{b}}) \ \& \ X(H^{\log}(\mathbf{c}^{\mathrm{b}})) \le X_{\min} \ \& \ c_{\min} < \max(\mathbf{c}^{\mathrm{b}}) \\ X(H^{\log}(\mathbf{c}^{\mathrm{b}})) & \text{otherwise} \end{cases}, \qquad (8)$$

where $H^{\log}(\mathbf{c}^{\mathrm{b}})$ and $H^{\mathrm{lin}}(\mathbf{c}^{\mathrm{b}})$ are calculated from Eqs. (6) and (7), respectively. We choose the minimum threshold values used in Eq. (8) as $X_{\min} = 10^{14}\,\mathrm{molecules/cm^2}$ and $c_{\min} = 0.1\,\mathrm{ppbv}$. In the remainder of the paper, we refer to this as the 'hybrid' method. Returning to the example shown in Fig. 2, the GEM-MACH column value at this retrieval location, $X(\mathbf{c}^{\mathrm{b}})$, is $5.37 \times 10^{15}\,\mathrm{molecules/cm^2}$, while the column value for the log-space operator $X(H^{\log}(\mathbf{c}^{\mathrm{b}}))$ is zero, so the hybrid method would use the observation operator that employs the linearized averaging kernel to compare this retrieval to the model. Had
the model profile not been 'zeroed-out', the original log-space averaging kernel would be used by the hybrid method, as no approximation to the averaging kernel would have been needed in that case.

The monthly frequencies in which the linear operator is chosen by the hybrid method are shown in Figure 3. Each panel shows the percentage of retrievals in a $0.5° \times 0.5°$ longitude/latitude bin in which the linear operator is chosen, while the percentage shown in the lower right corner of each panel is the percentage over the whole domain. In May, the log-space
observation operator is chosen for the model comparison for the majority of retrievals, with the linear observation operator being chosen for only 16.5% of retrievals, the majority of which are located in the Plains states/Prairie provinces and in the northeastern US and in eastern Canada. In contrast, the retrievals from June to August have the linear operator selected roughly 50% of time. For these months, most of the more remote northern locations in Canada and locations in the southern US and northern Mexico have the logarithmic operator selected, while locations in the rest of the US and Canada are more likely to
have the linear operator selected.



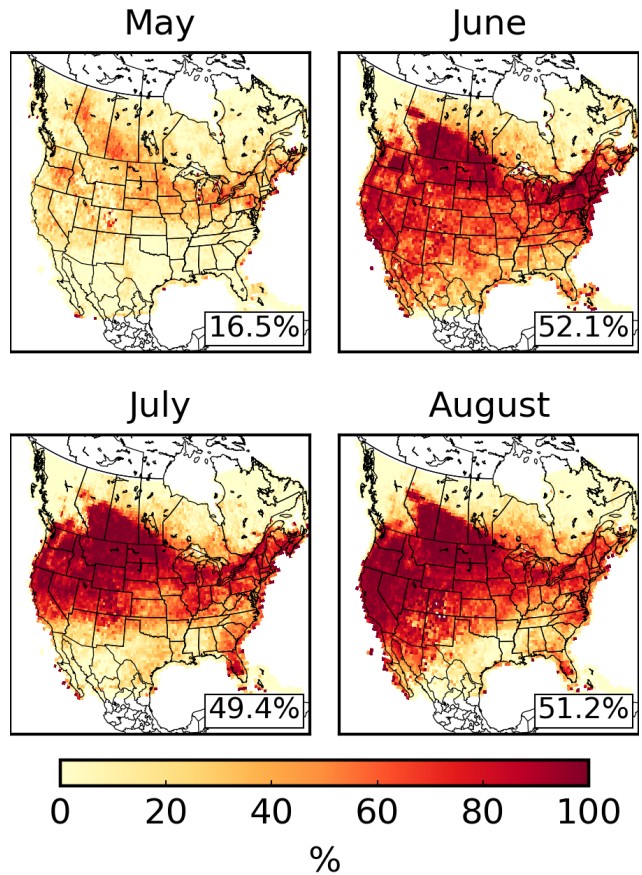

**Figure 3.** Fraction of the number of retrievals chosen to be compared with the model using the linearized averaging kernel in $0.5° × 0.5°$ longitude/latitude bins in the hybrid method. Percentages in the lower right corner show the percentage over the whole domain.

Lastly, to account for errors in the observation operator (from errors related to the issues discussed above as well as any other representation errors), the total column error variances as diagnosed by Shephard and Cady-Pereira (2015) were scaled by a factor of 100 to form the diagonal terms of **R** used in the inversions (similar to the regularization used in Cao et al. (2020)).

## 4 Results

We begin this section by discussing the monthly mean ammonia emissions produced by the inversion and its impact on the ammonia fields in GEM-MACH. We then examine the effect of using different observation operators on the inversion. Lastly, we determine the impact of the ammonia emissions inversion on PM formation, the PM size distribution, and the wet and dry deposition fields produced by GEM-MACH.



## 4.1 Ammonia Emissions Inversions

The ammonia total columns from the CrIS retrievals and their analogous quantities from the GEM-MACH model run using the original ammonia emission (the background) for May to August 2016 are shown in Figure 4, where the monthly mean column values within $0.5° × 0.5°$ longitude/latitude bins are shown for each month. Their differences (the innovations), are shown in the right column. In this figure, the GEM-MACH columns are derived using the hybrid observation operator of Section 3.3 and, unless stated otherwise, results in this section will be shown for comparisons using this observation operator. For the

innovation plots, each difference $i$ within a $0.5° × 0.5°$ longitude/latitude bin is weighted by $\mathbf{h}_i \mathbf{B}_{cc} \mathbf{h}_i^\mathrm{T} / (\mathbf{h}_i \mathbf{B}_{cc} \mathbf{h}_i^\mathrm{T} + \mathbf{R}_{i,i})$ in the computation of the mean for that bin, where $\mathbf{h}_i$ is the $i^\mathrm{th}$ row of the matrix $\mathbf{H}$, to approximate the weight given to that difference in the inversion (this roughly approximates the Kalman gain in observation-space). From the right column of Fig. 4, it is apparent that while the GEM-MACH background column values are larger than the CrIS retrievals in some places, such as the American Midwest, eastern North Carolina, and California's Central Valley in June and July, the CrIS retrieval columns

are larger than the GEM-MACH background values in most other locations within the domain. The largest differences between the CrIS retrieval columns and GEM-MACH background columns occur in the central continental US, from the American Midwest to northern Texas, California, and the Canadian prairie provinces.

The ammonia emissions inversions for May to August 2016 are shown in Figure 5 and the total change in emissions over the contiguous US and Canada are shown in Table 2. In Fig. 5 and subsequent figures within this section, the percentages shown in

the lower right corner of difference plots show the total difference over the model domain. Under the assumptions detailed in Section 3.1, the inversion attributes all differences between the GEM-MACH background values and the CrIS retrievals to the ammonia emissions. Accordingly, as the CrIS ammonia retrievals are larger than the GEM-MACH background values in most of the model domain, the inversion increases ammonia emissions in most places within the domain as well. The largest change in emissions occurs in May, where emissions over the domain increase by 41%, although the relative change in emissions in

Canada for June to August are larger than that for May. The spatial distribution of the emissions increment generally agrees with those found in Cao et al. (2020), who attributed the underestimations of emissions seen in much of the contiguous US to underestimations in emissions from both fertilizers and livestock (depending on location).

Although the inversions increase ammonia emissions in much of the domain, some notable exceptions to this are in southern Minnesota and Iowa, California's Central Valley, and eastern North Carolina for the months of June and July. These

correspond to areas with large agricultural industries (most significantly cattle in the Central Valley and swine in southern Minnesota/northern Iowa and eastern North Carolina) and represent significant emissions sources in the a priori emissions. The overestimations of emissions in June and July in these locations were also found by Cao et al. (2020) for 2014. While the inversions decrease major emissions sources in some months, in other months the emissions are increased at the same locations. For example, the inversion decreases emissions in much of southern Minnesota and Iowa in June and July, but increases emis-

sions in this area in May and August. A similar time-dependent behaviour is also seen for the Central Valley, North Carolina, and southern Saskatchewan. The time-dependence of the emissions increases/decreases in these locations may be due to the temporal profiles used in the processing of emissions inventories.





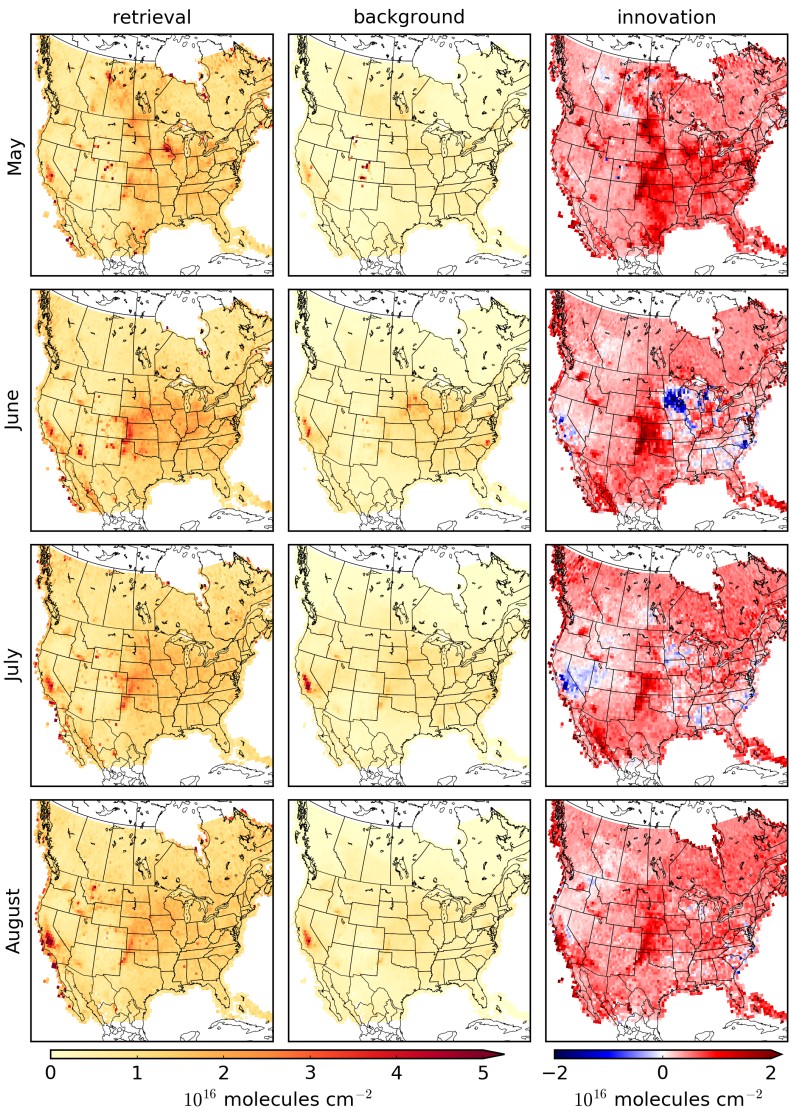

**Figure 4.** Monthly mean ammonia total columns for the CrIS retrievals (left column) as compared to the analogous GEM-MACH background values (middle column), as well as their difference (innovation, right column) for May to August 2016. The GEM-MACH background values are calculated using the hybrid method given in Eq. (8). The values plotted are monthly means within $0.5° \times 0.5°$ longitude/latitude bins. For plots of the innovation, each difference $i$ within a $0.5° \times 0.5°$ longitude/latitude bin is weighted by $\mathbf{h}_i \mathbf{B}_{cc} \mathbf{h}_i^{\mathrm{T}} / (\mathbf{h}_i \mathbf{B}_{cc} \mathbf{h}_i^{\mathrm{T}} + \mathbf{R}_{i,i})$ in the computation of the mean for that bin.

## 4.2 Impacts on Modeled Ammonia Surface Fields

With the new set of monthly mean ammonia emissions produced by the inversions, new hourly ammonia emissions were
generated (as described in Section 2.3) and used in updated GEM-MACH model runs while keeping the emissions of all other



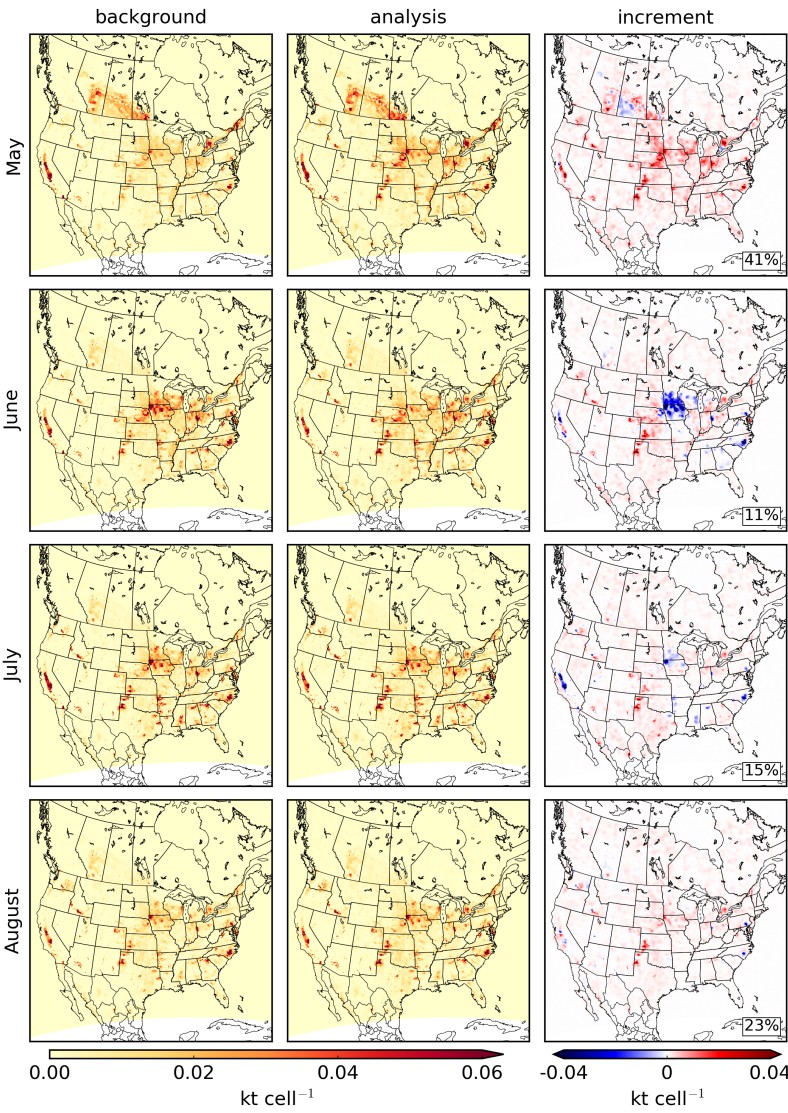

**Figure 5.** Monthly ammonia emissions (in kilotonnes cell$^{-1}$) for May to August 2016 on the $0.09° \times 0.09°$ model grid. The background emissions (left column) are those described in Section 2.3, the analysis (middle column) are the emissions produced by the inversions, and the increment (right column) is the difference between the two. For plots on the right, the percentages in the lower right corner show the change over the whole domain.

species unchanged. The changes in the predicted monthly mean ammonia surface volume mixing ratio (VMR) field between the original and updated GEM-MACH runs are shown in Figure 6(a), where the original monthly means and the changes to the means are shown in the left and right columns, respectively. As ammonia has a relatively short atmospheric lifetime, the changes to the ammonia surface VMR field closely resemble the changes to the ammonia emissions (shown in Fig. 5). As with





| month | contiguous US | | | Canada | | |
| | original (kt) | updated (kt) | change (%) | original (kt) | updated (kt) | change (%) |
|---|---|---|---|---|---|---|
| May | 315.87 | 460.46 | 45.78 | 140.73 | 179.58 | 27.61 |
| June | 456.57 | 477.56 | 4.60 | 48.38 | 76.94 | 59.02 |
| July | 425.98 | 461.91 | 8.43 | 41.06 | 67.73 | 64.94 |
| August | 335.30 | 392.69 | 17.11 | 45.05 | 73.06 | 62.18 |

**Table 2.** Total monthly ammonia emissions (in kilotonnes) for the original and updated sets of emissions for the contiguous US and Canada.

the emissions, the most significant changes in the ammonia surface VMR field occurs in the central US, California, and the Canadian prairie provinces. The largest increases in surface ammonia VMR occur in May, where values increase by as much ~6 ppbv in California's Central Valley and increase by 43% over the whole domain, while decreases of up to ~8 ppbv occur in July (in Central California as well).

Figure 7 shows the NMB values for ammonia observations from the NAPS and AMoN networks. In this figure, biases
are displayed for four different GEM-MACH runs, each of which use a different set of ammonia emissions: The original emissions and emissions from inversions using the hybrid, logarithmic, and linear observation operators. For the moment, we only consider the values for runs using the original emissions and inversions using the hybrid observation operator. The NMB values for these two cases, as well as the other statistics from Eq. (1), are shown in Table S1 of the Supplement. As seen in Fig. 7, when the original ammonia emissions are used, GEM-MACH underestimates surface ammonia as compared to both
the NAPS and AMoN networks. Using the ammonia emissions from the inversions that utilize the hybrid observation operator reduces this underestimation in every month for both networks. The largest reductions in NMB for both observation networks occurs in May, where the NMB is reduced by 19.1% and 25.1% for NAPS and AMoN, respectively. The reduction in bias is closer to 5–10% for June to August for both networks. We note that while the bias is reduced in all cases examined, some of the reductions in bias have lower statistical significance, such as for NAPS in June (see Table S1 of the Supplement).
The normalized standard deviation of differences and correlation coefficients for the surface ammonia observations are shown in Table S1 of the Supplement. From this table, we can see that while the standard deviation of differences decrease in most cases when the updated ammonia emissions are used (with NAPS in July and August being the exception), in all cases these changes have relatively low statistical significance. The correlation coefficients between GEM-MACH and the ammonia surface observations increase for all months for AMoN, but decreases for NAPS for the months of June, July, and August,
although the difference for June has low statistical significance.

The NMB values for each station are shown in Figure S3 of the Supplement. The NMB values of a majority of stations in the Central US improve when the updated emissions are used. However, a few clusters of stations near the US east coast have higher NMB values when the emissions from the inversions are used. In this data set, only one station is located in one of the regions that showed a significant drop in emissions (Minnesota/Iowa, North Carolina, the Central Valley, southern





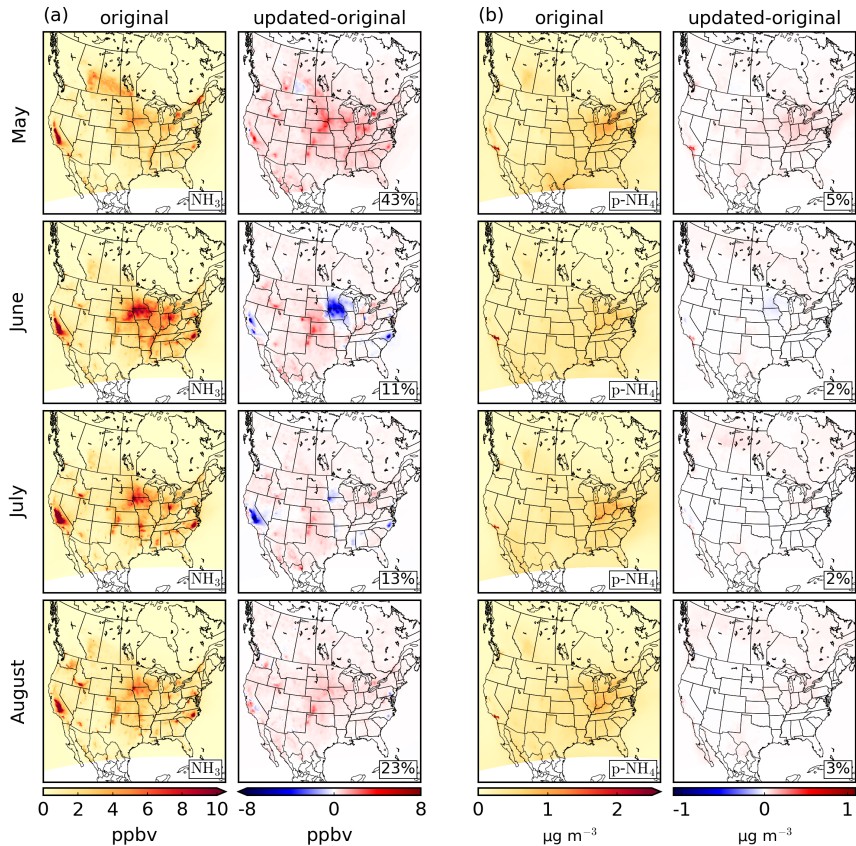

**Figure 6.** Monthly mean surface (a) NH$_3$ VMR and (b) p-NH$_4$ concentration fields from GEM-MACH for May to August 2016 on the $0.09° \times 0.09°$ model grid. In sub-figures (a) and (b), the left columns show the mean surface field when the original ammonia emissions are used and the right columns show the mean difference between the GEM-MACH runs with the updated ammonia emissions from the inversion and the original emissions. For plots in the right columns, the total difference over the model domain as a percentage of the original field is shown in the lower right corner. Plots for p-NH$_4$ show the total ammonium mass over both aerosol size bins.

Saskatchewan), which is located in Clinton, North Carolina. Observations from this station, which were only available in July and August, showed an over-prediction of ammonia for both of these months when the original emissions were used, as was also found for this location by Battye et al. (2019). Using the emissions from the inversions decreased the over-predictions at the Clinton station by 3.89 and 1.80 $\mu g\,\mathrm{m}^{-3}$ in July and August, respectively.

## 4.3   Observation Operator Selection

In Section 3.3, it was shown that the particular observation operator used for the model/retrieval comparison may have a significant impact on the emissions inversion. Figure 8 shows a comparison of the ammonia emissions increments that result from inversions that use the different observation operators described in Section 3.3. The increments shown in the middle and





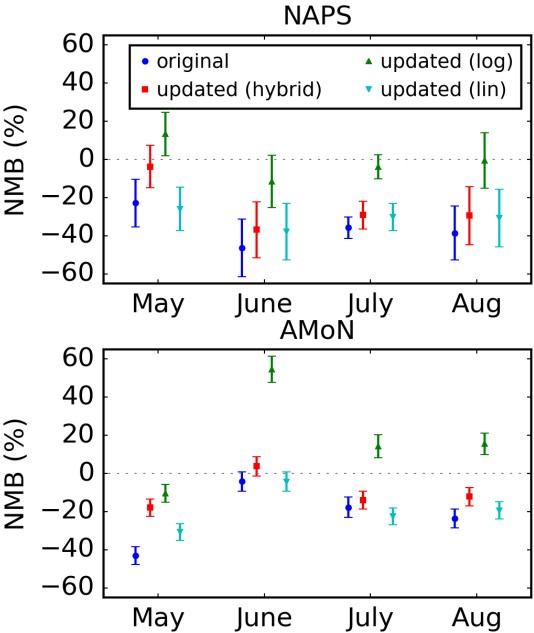

**Figure 7.** Normalized mean biases between the GEM-MACH ammonia surface fields and observations from the NAPS and AMoN networks for May to August 2016. In the legend, 'original' and 'updated' indicate the ammonia emissions used in GEM-MACH. The observation operator used in the inversion is indicated in the parentheses of the legend labels. Error bars denote the standard error.

right columns of Fig. 8 were the result of inversions in which all retrievals were compared to GEM-MACH using only the log-space or linear-space observation operators, respectively, while the left column shows the increment using the hybrid method

(identical to that in Fig. 5, shown for comparison).

The discussion in Section 3.3 suggested that if the log-space observation operator is used for the model comparison for all retrievals, the ammonia increment may be overestimated due to the 'zeroing out' of some model profiles. On the other hand, using the linear-space observation operator in all cases tends to overestimate the GEM-MACH analogue profile, thereby underestimating the increment. This general behavior is evident in the increments shown in Fig. 8, where the inversions that

use the log-space (linear-space) observation operator for all comparisons yield increments that are larger (smaller) in many locations than the increment from the hybrid operator. This is particularly notable in southern Saskatchewan in May and southern Minnesota and Iowa in June, where emissions were removed when the hybrid operator was used but had emissions added when the log-space operator was used instead. In contrast, when the linear-space observation operator was used, these regions had more emissions removed over a larger area. The largest differences between emissions increments occurs in June,

where the increment produced using the log-space observation operator in the mid-western US and California is significantly larger than that produced using the hybrid and linear-space operators.

Returning to Fig. 7, we can see that the choice of observation operator has a similar effect on the comparisons with ammonia surface observations. For the comparisons with the AMoN network, using the log-space operator raises the biases as compared





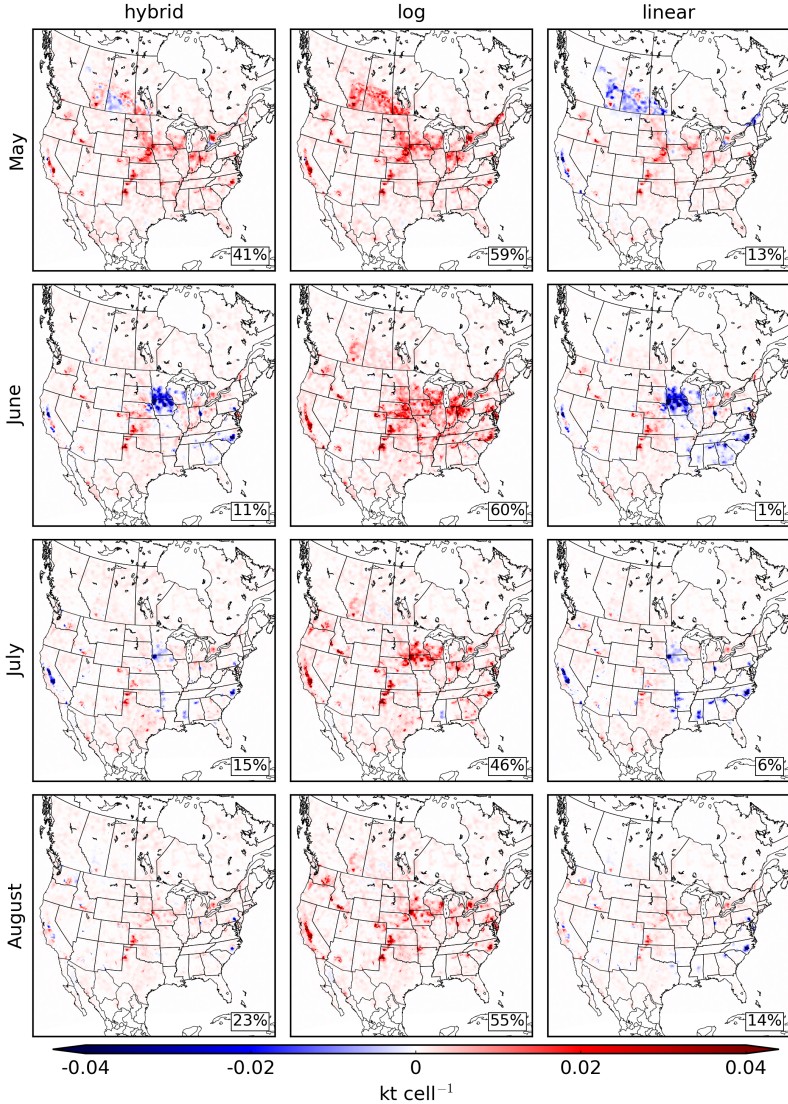

**Figure 8.** Monthly mean ammonia emissions increments (in kilotonnes cell$^{-1}$) for May to August 2016 on the $0.09° \times 0.09°$ model grid for inversions using different observation operators. Inversions displayed in the left, middle, and right columns use the hybrid, log, and linear observation operators, respectively. The left column, which is identical to the right column of Fig. 5, is shown for comparison. Percentages in the lower right corners show the change over the whole domain.

to the values using the hybrid observation operator, while using the linear-space observation operator lowers these values. The

same is true of NAPS for the month of May. However, for NAPS in June to August, there is little difference between the hybrid and linear-space operators since the hybrid method selects the linear-space operator the majority of the time in these months at the NAPS locations (seen by comparing Fig. 3 to Fig. S2 of the Supplement). In May, the hybrid method selects the log-space





operator most often at the NAPS and AMoN locations, resulting in larger differences between the hybrid and linear methods, where the NAPS and AMoN biases are about 23% and 13% more negative, respectively, when the linear-space operator is used

instead of the hybrid operator. Comparing the absolute NMB values of the linear-space and hybrid observation operators, the hybrid operator yields smaller or equal values in all cases.

Comparing the hybrid and log-space operators in Fig. 7, we can see that the method with the best performance as judged by the NMB changes with the month and network. The hybrid method yields the lowest bias for NAPS in May and for AMoN in June, July, and August, while the log-space method gives the lowest values for the remaining cases. The differences between

the hybrid and log-space operators are generally larger in June to August as compared to May, as the hybrid method selects the log-space operator for the majority of retrievals in May, in contrast for June to August where the log-space operator is selected less often (see Fig. 3). As noted above, the largest differences between the emissions increments produced using the hybrid and log-space observation operators occurs in June in the mid-western US and in California, where the log-space increment is significantly larger in these areas than for the hybrid increment. The impact of this large increment can be seen in Fig. 7, where

large overestimations of the GEM-MACH ammonia surface fields are seen in June for AMoN when the log-space operator is used in the inversion. The NMB is much larger for this case than for any other months and network, including cases that used the original ammonia emissions.

Overall, using the hybrid observation operator for the comparison of GEM-MACH to the CrIS retrievals in the inversion generally yields favorable results when evaluating the inversions using NAPS and AMoN observations. While using the log-

space operator yields good results for many cases, the large overestimation of ammonia emissions in some places in the US in June demonstrates that this operator must be used with caution. For the remainder of this paper, all results shown will be for inversions that use the hybrid operator.

### 4.4 Impacts on Predicted PM Formation

As ammonia plays an important role in inorganic PM formation, we now examine what impact, if any, the ammonia emissions

inversion has on species related to inorganic PM in GEM-MACH. To illustrate this process, Figure 9 shows the mean profiles for p-NH$_4$, p-NO$_3$, and p-SO$_4$ and the precursor gases NH$_3$, HNO$_3$, and SO$_2$ for May 2016 at two different locations. The panels on the left show the monthly mean profiles of GEM-MACH run with the original ammonia emissions, while the panels on the right show the mean differences between GEM-MACH run with the updated and original ammonia emissions. The upper and lower panels show the profiles near Toronto, Ontario and the city of Grand Junction in western Colorado, respectively. The

profiles for p-NH$_4$, p-NO$_3$, and p-SO$_4$ are computed by taking the sum of the fine and coarse aerosol bins of each species.

As the inversion increased the ammonia emissions in both locations shown in Fig. 9, the ammonia difference profile shown in the right panels are positive for both locations, peaking at the surface and decreasing with height. Fig. 9 shows little change in SO$_2$ or p-SO$_4$. The oxidation of S(IV) to S(VI) is primarily controlled by the aqueous-phase oxidation of S(IV) with ozone, hydrogen peroxide, and organic peroxides (Seinfeld and Pandis, 2006). While the S(IV) oxidation rate is sensitive to pH, and

thus may be sensitive to ammonia levels, in these locations this effect is small. Furthermore, since the formation of inorganic

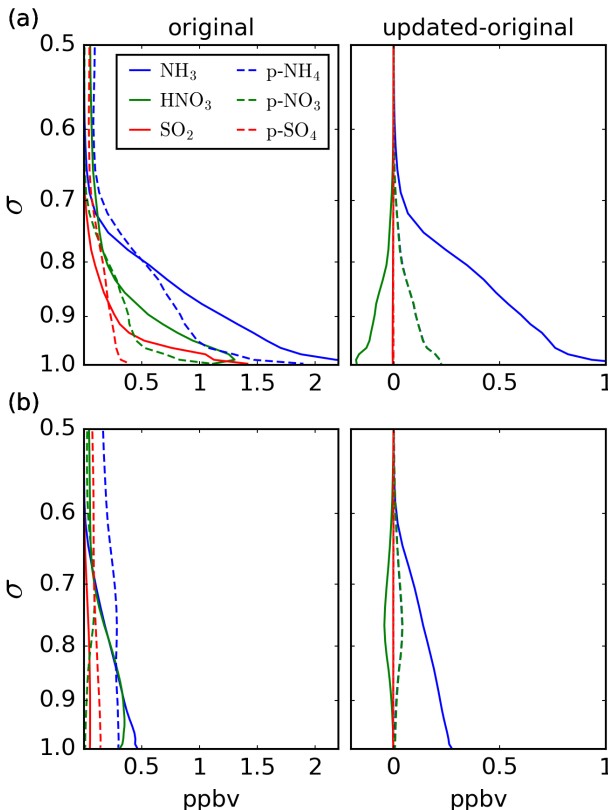

**Figure 9.** Monthly mean vertical profiles near (a) Toronto, Ontario and (b) Grand Junction, Colorado for May 2016. Plots on the left show mean profiles for GEM-MACH run with the original ammonia emissions, while plots on the right show mean differences between the updated and original runs. The vertical coordinate $\sigma$ is the ratio of the pressure to the surface pressure. The legend in the upper right plot applies to all plots.

PM through reactions such as $2NH_4^+(aq) + SO_4^{2-}(aq) \rightleftharpoons (NH_4)_2SO_4(s)$ does not change the total amount of sulfate, the total sulfate profile shows little change when the emissions from the inversions are used.

As ammonia preferentially neutralizes sulfate over nitrate, significant nitrate formation through the reaction $NH_3(g) + HNO_3(g) \rightleftharpoons NH_4NO_3(s)$ will only occur if ammonia levels exceed the amount needed to neutralize the sulfates. For the
profile near Toronto, the amount of ammonia close to the surface exceeds that required for sulfate neutralization, and so a non-negligible amount of nitrate formation can occur near the surface. This can been seen in the upper right panel of Fig. 9, which shows a decrease in $HNO_3$ that peaks at the surface and lessens with height, which is mirrored by increases in p-$NO_3$ and p-$NH_4$. In contrast, for the location in western Colorado shown in the lower panels of Fig. 9, at the surface ammonia levels are only high enough for sulfate neutralization, having a surface $NH_x$ (ammonia gas + ammonium aerosol) to sulfate ratio of
roughly half of that as for the location near Toronto. Consequently, little nitrate formation occurs at the surface for the location





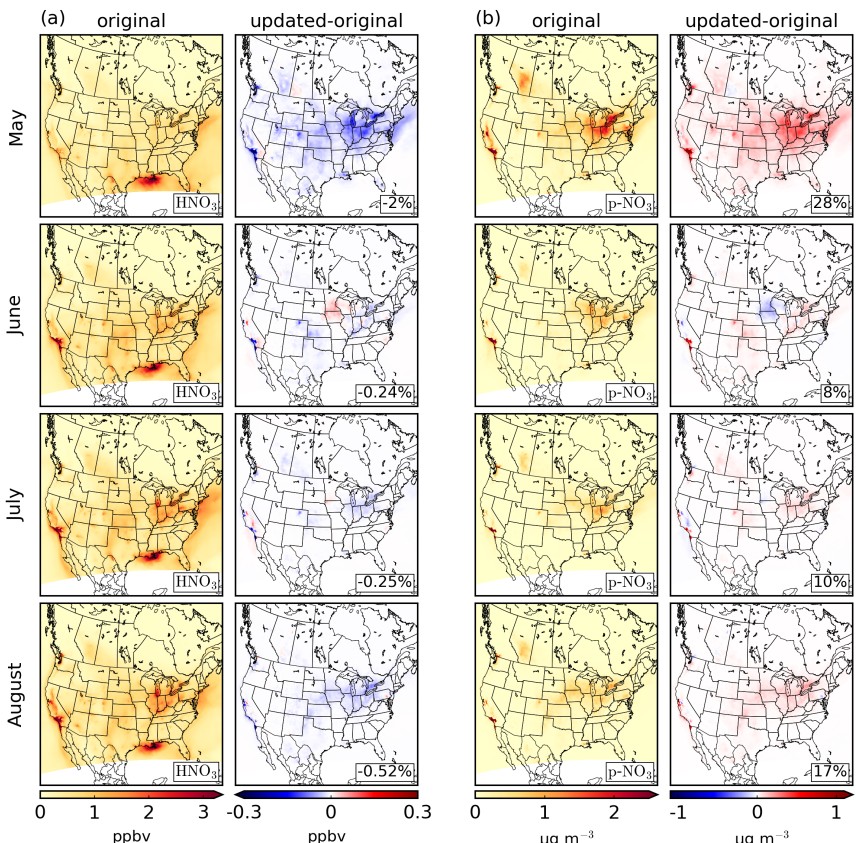

**Figure 10.** Same as Fig. 6, except for (a) HNO$_3$ and (b) p-NO$_3$.

in western Colorado. However, as the NH$_x$ to sulfate ratio increases with altitude at the western Colorado location, ammonium nitrate formation occurs at higher altitudes, as seen in the lower right panel of Fig. 9. As such, unlike the changes in ammonia, the largest changes in ammonium nitrate concentrations may or may not occur near the surface.

Figures 6(b) and 10 show the monthly mean differences in the GEM-MACH surface fields for p-NH$_4$, HNO$_3$, and p-NO$_3$
when the updated ammonia emissions are used. As seen in Fig. 10, areas with increased ammonia emissions have depleted levels of HNO$_3$ and increase levels of p-NO$_3$ due to increased neutralization with ammonia. As ammonia emissions increase in the majority of areas in the model domain, HNO$_3$ decreases and p-NO$_3$ increases in most regions within the domain. In southern California and the mid-western US, the monthly mean surface p-NO$_3$ levels increase by as much as $\sim$2 $\mu g\,m^{-3}$. The largest increases in surface nitrate occur in May, increasing by 28% over the whole domain. Note that the regions with
the largest increases in p-NH$_4$ and p-NO$_3$ do not necessarily coincide with the regions with the largest increases in ammonia emissions, as the spatial distribution of HNO$_3$ can have a large impact on the spatial distribution of ammonium nitrate.

The mean differences in the surface SO$_2$ and p-SO$_4$ GEM-MACH fields are shown in Figure S4 of the Supplement. The mean differences of the surface SO$_2$ and p-SO$_4$ fields range from -0.062–0.007 ppbv and -0.032–0.158 $\mu g\,m^{-3}$, respectively.



As previously noted by Makar et al. (2009) and noted above, these changes are in part due to changes in the aqueous-phase
oxidation rates of S(IV) to S(VI) caused by changes in pH, in addition to changes in the PM dry deposition rates due to changes
in the aerosol size distribution.

Figure 11 shows the NMB values for $HNO_3$, p-$NH_4$, and p-$NO_3$ observations from the CAPMoN and CASTNET networks
for the GEM-MACH runs with the original ammonia emissions and the emissions from the inversions. Statistics for these
species, as well as for $SO_2$ and p-$SO_4$, are also given in Tables S1–S3 of the Supplement. In addition, statistics for individual
stations are shown in Figures S5–S7 of the Supplement. As seen in Fig. 11, when the original ammonia emissions are used,
GEM-MACH over-predicts surface $HNO_3$ and under-predicts p-$NH_4$ and p-$NO_3$. These over-/under-predictions are seen at
almost every station (see Figs. S5–S7 of the Supplement). The magnitude of the CAPMoN NMB values for $HNO_3$, as well as
the CASTNET NMB values for $HNO_3$ in August, are in general larger than the magnitudes of the NMB values for p-$NH_4$ and
p-$NO_3$, with the $HNO_3$ NMB value reaching as high as $\sim$170% for CAPMoN in August. The biases for all p-$NH_4$ and p-$NO_3$
cases, as well as CASTNET $HNO_3$ for May to July and CAPMoN $HNO_3$ for May, are within 100%. When the ammonia
emissions from the inversions are used, ammonium nitrate production increases in most locations within the model domain,
resulting in decreases in the overestimations of $HNO_3$ and in the underestimations of p-$NH_4$ and p-$NO_3$. This improvement
is most notable in May, where the p-$NH_4$ and p-$NO_3$ biases improve by $\sim$5% and $\sim$10–13%, respectively. While the NMB
values for $HNO_3$ are reduced when the updated ammonia emissions are used, these reductions have low statistical significance
for most cases and are relatively small as compared to the overall size of these biases, which range between 50% and 170%.
The overestimation in $HNO_3$ will be discussed further in Section 4.5. As the changes in the $SO_2$ and p-$SO_4$ surface fields were
small, the change in the biases of the corresponding observations were small as well (see Table S3 of the Supplement).

Although using the updated ammonia emissions increased the NSTD values for $HNO_3$, p-$NH_4$, and p-$NO_3$ in some cases,
in all cases the change in NSTD had low statistical significance (see Tables S1 and S2 of the Supplement). Likewise, the
changes in correlation coefficients for these observations had relatively low statistical significance (with the possible exception
of CAPMoN p-$NH_4$ in May).

Both CAPMoN and CASTNET employ a 3-stage filter pack consisting of, in series, a Teflon filter to collect p-$NH_4$, p-$NO_3$,
and p-$SO_4$, followed by a nylon filter to collect $HNO_3$ (and some $SO_2$), and finally cellulose filters impregnated with $K_2CO_3$
to collect the remaining $SO_2$ (see Table 1). The volatilization of ammonium nitrate on Teflon filters can be significant (Hering
and Cass, 1999; Ashbaugh and Eldred, 2004), especially during warm months. For these types of filter packs, the volatilization
of aerosol ammonium nitrate to gaseous $NH_3$ and $HNO_3$ on the Teflon filter and the subsequent collection of $HNO_3$ on the
nylon filter can cause an underestimation of nitrate and overestimation of $HNO_3$ (Sickles and Shadwick, 2002, 2007, 2015).
Accounting for this effect would make the biases shown in Fig. 11 larger, but the decrease in the (unnormalized) biases when
using the emissions from the inversion would remain the same. In contrast, coarse particles collected on the Teflon filter may
scavenge $HNO_3$, which would create a high nitrate bias and a low $HNO_3$ bias (Sickles and Shadwick, 2007, 2015).





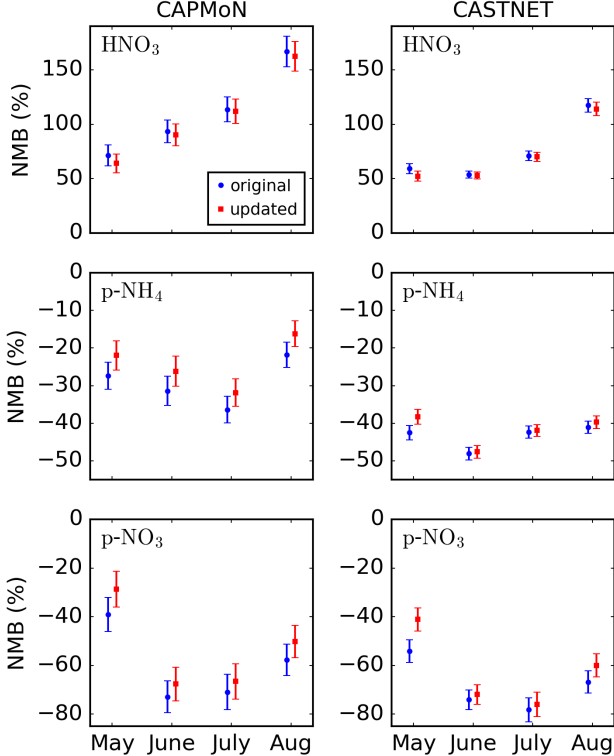

**Figure 11.** Normalized mean biases between GEM-MACH $HNO_3$, p-$NH_4$, and p-$NO_3$ surface fields and observations from the CAPMoN and CASTNET networks for May to August 2016. In the legend, 'original' and 'updated' indicate the ammonia emissions used in GEM-MACH. Error bars correspond to the standard error.

## 4.5 Impact on PM Size Distribution

The previous section examined the impact of the ammonia emissions inversions on the total amount of ammonium, nitrate, and sulfate. In this section, we explore the impact of the emissions inversions on the size distribution of inorganic PM, and focus on the impact on $PM_{2.5}$ in particular due to its significance in air quality. As mentioned in Section 2.3, GEM-MACH

first uses the HETV package to determine the gas/particle partitioning of inorganic species using a bulk approach, followed by the partitioning of the PM into size-resolved bins. As such, much of the focus of this section will be on the partitioning into size bins following the bulk calculation. In the previous section, it was shown that using the ammonia emissions from the inversions increased ammonium nitrate production in most locations within the model domain (with a few notable exceptions mentioned in previous sections). While significant amounts of nitrate can be found in both fine and coarse particles, ammonium

nitrate production is usually associated with fine particles, whereas the nitrate found in coarse particles more often results from reactions between $HNO_3$ and crustal materials (calcium, potassium, magnesium) or sodium chloride (Seinfeld and Pandis, 2006).





Figure 12 shows the monthly mean change in the fine ($PM_{2.5}$) and coarse-fraction ($PM_{2.5-10}$) ammonium and nitrate GEM-MACH fields when the ammonia emissions from the inversions are used. One of the most notable features of these plots are the large amounts of ammonium and nitrate being added to the coarse-fraction bin. This is particularly notable for ammonium, where for many locations where total ammonium increases (see Fig. 6), the amount of mass added to the coarse-fraction bin is larger than that added to the fine bin, and in some locations the mass in the fine bin actually decreases.

GEM-MACH uses a quasistationary bin-size structure (Jacobson, 1997), where particle volumes are initially allowed to grow or shrink, but are then fit back into a stationary bin structure following the change in particle volume. After the heterogeneous bulk chemistry is run in HETV, almost all of the additional ammonium and nitrate is added to particles in the fine bin. However, due to the use of only two size bins, in many cases the fitting done by the quasistationary structure reallocates large amounts of mass to the coarse-fraction bin, sometimes moving mass from the fine to the coarse-fraction bin by amounts that exceed the initial amount added to the fine bin. This is a clear limitation of the 2-bin model compared to one with a larger number of size bins.

The NMB values between the GEM-MACH $PM_{2.5}$-$NO_3$ and $PM_{2.5}$-$NH_4$ fields and observations from the NAPS, CSN, and IMPROVE networks are shown in Figure 13. Before examining the impact of the emissions inversions on $PM_{2.5}$, we begin by comparing the NMB values for the total ammonium and nitrate observations (Fig. 11) to the observations of just the $PM_{2.5}$ component (Fig. 13) for the GEM-MACH runs that use the original set of emissions. We can see that while p-$NH_4$ and p-$NO_3$ were underestimated for all cases shown, $PM_{2.5}$-$NH_4$ and $PM_{2.5}$-$NO_3$ are overestimated for NAPS and CSN. The relative magnitude of the overestimations of $PM_{2.5}$-$NH_4$ and $PM_{2.5}$-$NO_3$ for NAPS and CSN are much larger than the magnitude of the underestimations of p-$NH_4$ and p-$NO_3$, with a number of the $PM_{2.5}$-$NH_4$ and $PM_{2.5}$-$NO_3$ NMB values exceeding 100%. In contrast, the NMB values for $PM_{2.5}$-$NO_3$ from IMPROVE have significantly smaller absolute values and GEM-MACH underestimates the fields in many cases.

Now turning our attention to the impact of the emissions inversions on the biases, we can see that in most cases using the emissions from the inversion degrades the NMB values, though for many cases the size of this degradation is much smaller than the magnitude of the overall bias. Furthermore, if less mass was transferred from the fine to the coarse ammonium and nitrate bins during the quasistationary fitting procedure, then the biases for NAPS and CSN $PM_{2.5}$-$NH_4$ and $PM_{2.5}$-$NO_3$ would increase further.

Before discussing potential biases in the GEM-MACH model, we consider possible biases in the observations of $PM_{2.5}$-$NH_4$ and $PM_{2.5}$-$NO_3$. For the nylon filters used in the CSN and IMPROVE networks, the nitrate loss from the volatilization of ammonium nitrate was estimated to be minor (Yu et al., 2005) as the nylon filter can recapture volatilized $HNO_3$, although loss of ammonium can be significant (Yu et al., 2006). In the NAPS network, nitrate is collected on Teflon filters, on which the volatilization of ammonium nitrate can be significant, as discussed in Section 4.4. Some NAPS stations mitigate this nitrate loss through the use of a nylon filter downstream of the Teflon filter that can collect the volatilized $HNO_3$. For this reason, NAPS stations without the downstream nylon filter were not used in this work.

Many air quality models have been shown to overpredict the concentration of fine nitrate and/or ammonium (although many underpredict fine nitrate in California), often with biases larger in magnitude than biases for sulfate (Yu et al., 2005; Heald





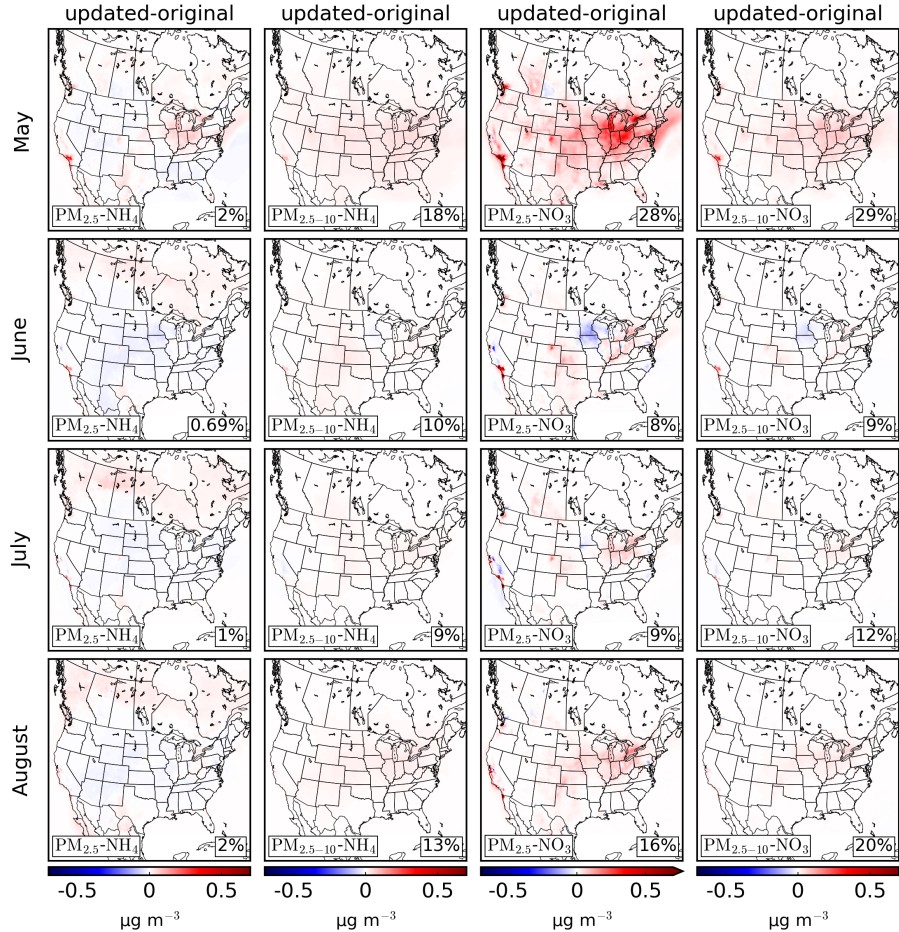

**Figure 12.** Monthly mean differences of GEM-MACH surface fields for fine ($PM_{2.5}$) and coarse-fraction ($PM_{2.5-10}$) ammonium and nitrate for May to August 2016 on the $0.09° \times 0.09°$ model grid. Plots show mean differences between the run with the updated ammonia emissions and the original emissions. The total difference over the model domain as a percentage of the original field is shown in the lower right corner of each plot.

et al., 2012; Walker et al., 2012). While only two PM size bins were used in this work, which very likely plays a significant role in the difficulty predicting fine ammonium and nitrate as noted above, significant overpredictions of $PM_{2.5}$ ammonium and/or nitrate have also been observed in other models that use significantly more PM size bins (Trump et al., 2015; Zakoura and Pandis, 2018).

Having found similar overestimations of nitric acid and fine nitrate in the GEOS-CHEM model using the ISORROPIA II model (Fountoukis and Nenes, 2007) for the inorganic gas/particle phase partitioning, Heald et al. (2012) found that if the nitric acid levels were artificially lowered by 75%, then the overprediction of fine nitrate decreases significantly and greatly improved the agreement with $PM_{2.5}$-$NO_3$ observations from the IMPROVE network. A decrease of nitric acid levels of this magnitude





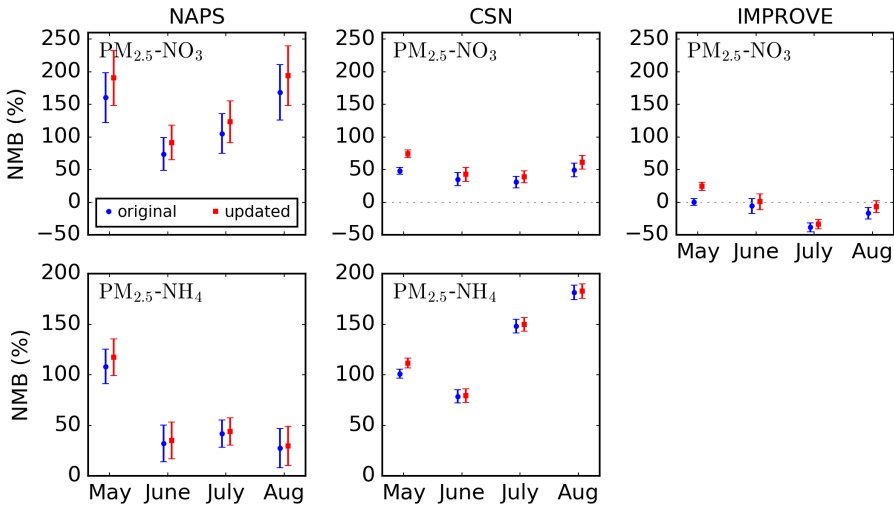

**Figure 13.** Monthly normalized mean biases between GEM-MACH $PM_{2.5}$-$NO_3$ and $PM_{2.5}$-$NH_4$ surface concentration fields and observations from the NAPS, CSN, and IMPROVE networks for May to August 2016. In the legend, 'original' and 'updated' indicate the ammonia emissions used in GEM-MACH. Error bars correspond to the standard error.

would also significantly decrease the high biases in the GEM-MACH nitric acid fields with observations from the CAPMoN and CASTNET networks seen in Fig. 11. However, the mechanism for a decrease of this magnitude was not identified. Heald et al. (2012) and Walker et al. (2012) both showed that significantly reducing the nighttime nitric acid production via the reaction $N_2O_5 + H_2O \rightarrow 2HNO_3$ in GEOS-CHEM only yields modest reductions in the $PM_{2.5}$-$NO_3$ bias. Similarly, Heald

et al. (2012) found that changes to the oxidation rate of $NO_2$ by OH, the OH concentration, temperature, and humidity in GEOS-CHEM are unlikely to yield $HNO_3$ decreases large enough to significantly reduce the $PM_{2.5}$-$NO_3$ biases observed (although these factors may partially explain the $PM_{2.5}$-$NO_3$ biases). Zakoura and Pandis (2018) demonstrated that coarser horizontal grid resolutions may artificially dilute $NO_x$ plumes, resulting in higher $N_2O_5$, $HNO_3$, and nitrate levels, although while using finer grid resolutions lessened the overproduction of fine nitrate, significant biases remained.

The competition of ammonia with crustal-material base cations and sea-salt for reaction with $HNO_3$ can significantly impact the size distribution of nitrate. Reactions of $HNO_3$ with crustal materials and sea-salt are currently not implemented in HETV. While including these reactions in HETV may significantly change the nitrate size distribution, the overprediction of fine nitrate has been shown to persist in models using ISORROPIA II, which includes these reactions (Walker et al., 2012; Trump et al., 2015). Trump et al. (2015) demonstrated that heterogeneous partitioning models that are based on the assumption of

thermodynamic equilibrium, such as ISORROPIA II (and HETV), may not accurately represent the interactions between nitric acid and coarse particles and that using an explicit mass transfer approach for these interactions can shift nitrate from the fine to the coarse mode in locations with significant amounts of sea-salt or dust.

Figure 14 shows the NMB values for total $PM_{2.5}$ for observations from NAPS and AQS networks for BAM and TEOM instruments. The total $PM_{2.5}$ is underestimated by GEM-MACH, with NMB values in the range of -60% to -2%. Although





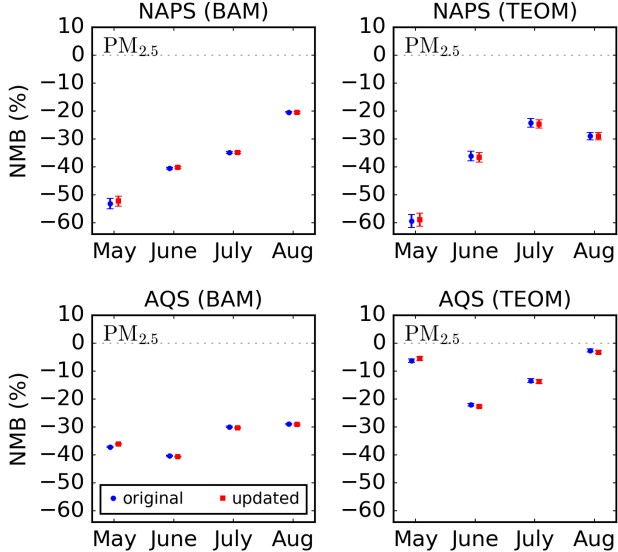

**Figure 14.** Normalized mean biases between GEM-MACH total $PM_{2.5}$ surface fields and BAM and TEOM observations from the NAPS network and observations reported to the AQS for May to August 2016. In the legend, 'original' and 'updated' indicate the ammonia emissions used in GEM-MACH. Error bars correspond to the standard error.

$PM_{2.5}$-$NH_4$ is overestimated, and $PM_{2.5}$-$NO_3$ is overestimated for the CSN and NAPS networks (Fig. 13), large underestimations of fine organic PM (Stroud et al., 2011) counteract the overestimations in fine inorganic PM. Only small changes are seen in the total $PM_{2.5}$ biases when the emissions from the inversions are used. This is in part due to many locations where GEM-MACH predicts increases in $PM_{2.5}$-$NO_3$ also predicts decreases in $PM_{2.5}$-$NH_4$.

### 4.6 Impact on Deposition and Critical-Load Exceedances

The previous sections showed that using the ammonia emissions from the inversions led to non-negligible changes in the levels of ammonia, ammonium, and nitrate ambient concentrations. In this section, we examine the changes in the removal of these species through deposition. Figure 15 shows the spatial distribution of monthly changes in the total (dry + wet) deposition rates of $NH_x$ and $HNO_3 + NO_3$. The changes in $NH_x$ deposition closely resembles the changes in the surface fields shown in Fig. 6. The largest increases in deposition were in May, where $NH_x$ total deposition increased by as much as $3.7 \times 10^{-4}$ moles m$^{-2}$ day$^{-1}$. The largest decreases in deposition were in the Minnesota/Iowa region in June, where deposition dropped by as much as $3.4 \times 10^{-4}$ moles m$^{-2}$ day$^{-1}$. The changes in $HNO_3 + NO_3$ deposition are an order of magnitude less than the changes in $NH_x$ deposition. While the changes in ammonium and nitrate were similar in magnitude, the changes in $NH_3$ were much higher than the changes in $HNO_3$, leading to much larger changes in the amount of $NH_3$ scavenged by precipitation as compared to $HNO_3$. As with $NH_x$, the largest increases in $HNO_3 + NO_3$ deposition occur in May, but only increase by as much as $0.53 \times 10^{-4}$ moles m$^{-2}$ day$^{-1}$.





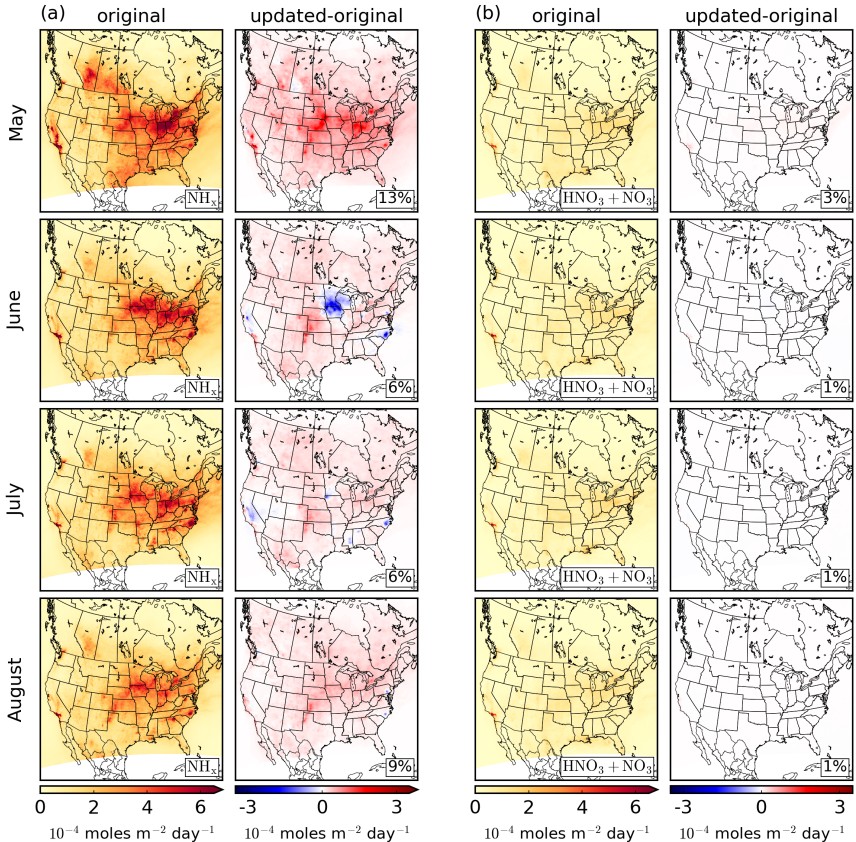

**Figure 15.** Same as Fig. 6, except for the total (dry + wet) deposition rates of (a) $NH_x$ and (b) $HNO_3 + NO_3$.

Figure 16 shows the NMB values for ammonium precipitation observations from the CAPMoN, NTN, and AIRMoN precipitation-chemistry networks. The full set of statistical measures for the precipitation-chemistry observations are given in Tables S4 and S5 of the Supplement and statistics for each station are shown in Figure S8 of the Supplement. From Fig. 16, we can see that GEM-MACH under-predicts ammonium wet deposition in all cases, with NMB values ranging between

-80% and -50%. As with the under-prediction of p-$NH_4$, the under-prediction ammonium wet deposition is seen at almost every station (see Fig. S8 of the Supplement). Using the updated ammonia emissions reduces these biases, with reductions as large as 8% occurring in May. The reductions for June to August are less statistically significant than the reductions in May. GEM-MACH underestimates nitrate wet deposition as well, although the updated ammonia emissions have little effect on these biases. The changes in NSTD and correlation coefficient generally have low statistical significance, with the possible

exception of the correlation coefficient for ammonium NTN in June (see Tables S4 of the Supplement).

The *critical load* of an ecosystem refers to an estimate of the maximum exposure of an ecosystem to a pollutant in which damage does not occur to the ecosystem. In the present context, the critical load refers to the maximum capacity of an ecosystem to absorb acidifying deposition without harming the ecosystem. The critical-load exceedance is equal to the total acidifying





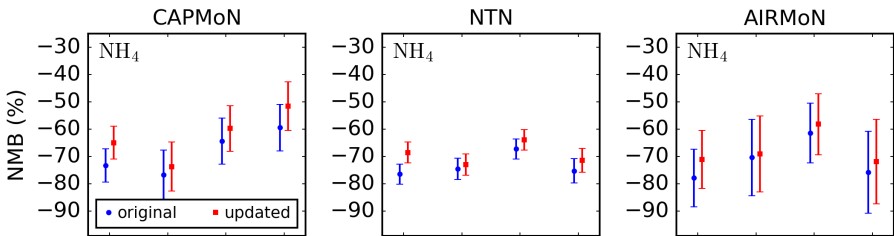

**Figure 16.** Normalized mean biases of GEM-MACH with original and updated ammonia emissions as compared to ammonium precipitation observations from the CAPMoN, NTN, and AIRMoN networks. Error bars denote the standard error.

deposition minus the critical load, so that a positive critical-load exceedance value implies that damage may be done to the

ecosystem. In the remainder of this section, we examine the impact of the emissions inversions on the critical-load exceedances for upland forests in Canada.

We use a steady-state model for the estimation of critical loads, which computes critical-load values assuming that the ecosystem has reached a chemical steady-state. Exceedances of these critical loads therefore do not indicate that damage will be done to the ecosystem immediately, but that damage will occur some time in the future when the steady state is reached. The

lag time between initial critial-load exceedance and actual damage to the ecosystem can be computed using dynamic models (CLRTAP, 2007). Critical-load exceedances are computed using the total (wet + dry) deposition from acidifying sulfur and nitrogen species, including $NH_x$ which can become acidifying through nitrification (Sverdrup and De Vries, 1994; Jeffries and Ouimet, 2005). Critical-load values for upland forests were estimated using the model of Ouimet (2005) (see also Makar et al. (2018)).

As inversions were only done for May to August, annual deposition values for the updated emissions case were calculated using deposition fields for September to April from a GEM-MACH run that used the original ammonia emissions. As only four months differ between the two cases, the changes in annual deposition may be underestimated as compared to the case where inversions are performed for all months of the year. However, as ammonia emissions in Canada are generally much lower during the late fall and winter, the change in emissions in the spring and summer are expected to be much larger than those in

colder months. Therefore, we expect that the annual deposition values using updated emissions for only May to August will capture the majority of the change in the annual values.

The annual critical-load exceedances using the original ammonia emissions are shown in Figure 17(a), which displays the critical-load exceedance as a percentage of the critical load. We see that many regions exceed the critical load (areas in red), including a large portion of Ontario and Quebec. Fig. 17(b) shows the change in total annual S + N deposition, as a percentage

of the critical load, when the ammonia emissions from the inversions are used. From Fig. 17(b), we see that the emissions inversions increased the annual deposition throughout forested regions in Canada, with increases of up to 19% of the critical-load value.



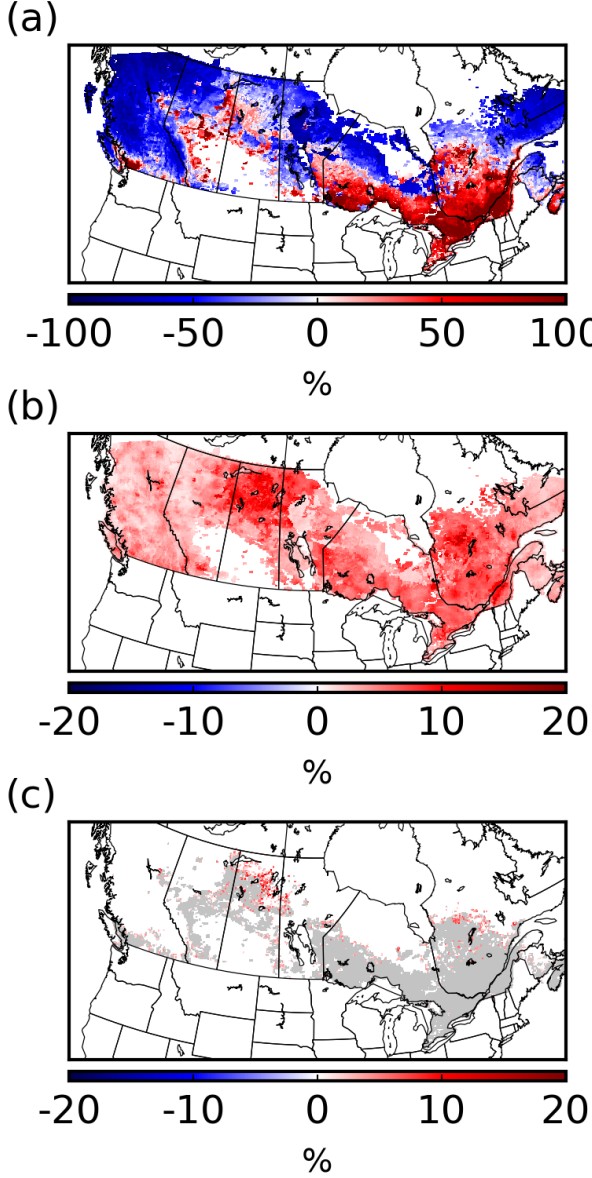

**Figure 17.** Annual critical-load exceedances on the $0.09° \times 0.09°$ model grid. Panels (a) and (c) show the exceedances using the original and updated ammonia emissions, respectively. Panel (b) shows the annual increase in total S + N deposition from the inversions. In panel (c), only areas in excess using the emissions from the inversion but not in excess using the original emissions are plotted with the colour scale, while areas that are in excess in both cases are shown in grey. All values are displayed as a percentage of the critical-load values.

The critical-load exceedance when the emissions from the inversions are used are shown in Fig. 17(c), again displayed as a percentage of the critical-load values. In this plot, areas that are in excess when either emissions are used are shown in grey to highlight and areas that are not in excess are not show. Only areas that are were not in excess using the original emissions






but are in excess when the emissions from the inversions are used are shown using the color scale to highlight the additional areas exceeding the critical load. Using the emissions from the inversions expands the original areas of exceedance, of total area $4.385 \times 10^7$ km$^2$, by $1.3 \times 10^5$ km$^2$. Although this only represents a change of about 0.29% in the area, from Fig. 17(b), we see that regions that were already deemed to exceed the critical load using the original emissions may become even more
acidifying, with increases in annual deposition increasing by 10-20%.

## 5    Conclusions

As ammonia emissions are increasing or staying constant in many parts of North America, in contrast to other pollutants that have decreasing trends, ammonia is increasingly becoming a species of concern. However, current bottom-up ammonia emissions inventories are highly uncertain, especially concerning the spatial and temporal distribution of emissions. Recently
developed ammonia retrievals from satellite-borne instruments such as CrIS provide observations of atmospheric ammonia with global coverage. In this study, we developed an ensemble-variational emissions inversion system that utilizes CrIS ammonia retrievals to refine estimates of ammonia emissions that are used in the Canadian GEM-MACH air quality forecasting model. The use of an ensemble allows for the advection, deposition, and chemical reactions of ammonia to be modeled by GEM-MACH within the inversion without requiring an adjoint model. To mitigate the relatively large uncertainties on individual
retrievals, each inversion used a month's worth of retrievals to produce new estimates of the monthly mean ammonia emissions on a grid with a horizontal spacing of $0.09°$ ($\sim$10 km).

To examine the performance of the inversion system, inversions for May to August 2016 were performed, covering some of the months when ammonia levels within the model domain are at their highest. For these months, the CrIS retrievals were larger than the analogous quantities in GEM-MACH for most locations within the model domain, although the opposite was true for
some regions such as parts of California, Minnesota, and Iowa in some months. Consequently, the emissions inversions added ammonia emissions in most of the model domain. The largest increases in emissions from the inversions were in May, with emissions increasing by 45.8% in the contiguous US and by 27.6% in Canada. The emissions increases in May were largest in the central US, California, and the Canadian Prairies. The largest emissions decreases were in June and July in Minnesota, Iowa, North Carolina, and California. When GEM-MACH was rerun with the ammonia emissions derived from the monthly
mean emissions produced by the inversions, the largest changes in the GEM-MACH surface ammonia field occurred in May, where its monthly mean increased by as much as $\sim$7 ppbv in some locations and increased by 43% over the whole model domain. The inversions also decreased the emissions significantly in some regions for the months of June and July, resulting in decreases by as much as $\sim$9 ppbv in monthly mean surface ammonia in these locations. Using the ammonia emissions produced by the inversions reduced the underestimation of the GEM-MACH ammonia surface field in comparison to observations from
the NAPS and AMoN networks, with the largest reductions in NMB occuring in May, where biases were reduced by 19.1% and 25.1% for the NAPS and AMoN networks, respectively.

A novel hybrid approach for comparing logarithmic retrieval parameters to a model profile in an inversion was developed that attempts to pick the best method of comparison between utilizing the averaging kernel in log-space or linear-space de-





pending on the situation. Inversions that used the log-space (linear-space) averaging kernel for all comparisons produced
emissions increments that were significantly more positive (negative) as compared to the increments produced using the hybrid method. When comparing the resulting GEM-MACH ammonia surface fields to observations from the NAPS and AMoN networks, the hybrid comparison method yielded smaller or equal biases than when using the linear-space operator. For some months/networks, using the log-space operator for all comparisons yielded biases lower in magnitude than that produced using the hybrid method. However, the log-space operator also yielded the largest NMB value for all cases examined, at just under
60%. Overall, the hybrid method developed in this work was effective in reducing the biases between GEM-MACH and the ammonia surface observations.

The ammonia emissions inversions increased ammonium nitrate production in GEM-MACH over much of the model domain. The largest increases occurred in May, where surface nitrate increased by 28% over the model domain. In some regions, such as southern California and the mid-western US, the monthly mean surface nitrate increased by as much as 2 $\mu g\,m^{-3}$.
Using the original set of ammonia emissions, GEM-MACH over-predicted $HNO_3$ and under-predicted p-$NH_4$ and p-$NO_3$ as compared to observations from the CAPMoN and CASTNET networks. Using the ammonia emissions from the inversions reduced these over-/under-predictions, although the reductions in the biases of $HNO_3$ were relatively small compared to the overall bias values. However, significant reductions in the biases of p-$NH_4$ and p-$NO_3$ were seen in many cases, most notably for p-$NO_3$ in May where biases were reduced by 10–13%.

Using the ammonia emissions from the inversions increased $NH_x$ deposition in most regions within the model domain. The change in deposition was greatest in May, where the deposition over the domain increased by 13%. Using the emissions from the inversions reduced biases with ammonium precipitation observations from the CAPMoN, NTN, and AIRMoN networks, with reductions as large as 8% in May. From an examination of the critical-load values for upland forests in Canada, the increase in ammonium deposition from the inversions added $1.3 \times 10^5$ $km^2$ to areas in danger of future ecosystem harm and
added 10-20% more potentially acidifying deposition to areas already designated as in risk of future harm.

Although using the ammonia emissions from the inversions improved the agreement between surface observations of total ammonium and nitrate, these improvements were not seen when compared to speciated $PM_{2.5}$ observations. Using the currently operational version of GEM-MACH, which only has 2 aerosol size bins, likely played a large part in this result. However, other studies suggest that this may not be the only cause of this result, and other factors such as the interactions of ammonia
with crustal materials may need to be accounted for in the heterogeneous chemistry code. Although using the emissions from the inversions degraded the agreement between GEM-MACH and some of the speciated $PM_{2.5}$ surface observations, little impact was seen when comparing to total $PM_{2.5}$ observations.

This work focused on improving the performance of the currently operational version of GEM-MACH. As the inversions in this study used a whole month's worth of retrievals to yield monthly mean emissions, one potential operational implementation
would be to use inversion results for the current month from the year prior. Although this creates a lag of one year, typical lags in operational emissions inventories are much larger than a year. Alternatively, an operational implementation could involve inversions being performed periodically with a moving time window using retrievals from the current year to shorten this time lag. However, to properly resolve temporal features in the ammonia signal such as sharp increases due to the application of





fertilizer, the inversion time window may have to be reduced, so for example inversions using the last 1–2 weeks worth of
data could produce updated emissions for the current week. Although the currently operational version of GEM-MACH does
not include a bidirectional flux model for ammonia, the impact of the inversions on a research version of GEM-MACH that
includes a bidirectional flux model will be the subject of future work.

## Appendix A: Details of the Ensemble-Variational Method for Emissions Inversions

In this appendix, we review in detail the ensemble-variational method as applied to emissions estimation. We represent the
model by an operator $\mathcal{M}$ that relates the emissions $\mathbf{e}$ to atmospheric concentrations $\mathbf{c}$ by $\mathbf{c} = \mathcal{M}(\mathbf{e}, \boldsymbol{\theta})$, where the vector $\boldsymbol{\theta}$
represents all model input parameters other than the emissions. If the combined emissions/concentration model state vector $\mathbf{x}$
is represented by

$$\mathbf{x} = \begin{bmatrix} \mathbf{c} \\ \mathbf{e} \end{bmatrix}, \tag{A1}$$

then the background error covariance $\mathbf{B}$ can be represented in block form as

$$\mathbf{B} = \begin{bmatrix} \mathbf{B}_{cc} & \mathbf{B}_{ce} \\ \mathbf{B}_{ec} & \mathbf{B}_{ee} \end{bmatrix}, \tag{A2}$$

where $\mathbf{B}_{ec} = \mathbf{B}_{ce}^{\mathrm{T}}$. In the expression above, $\mathbf{B}_{cc}$ and $\mathbf{B}_{ee}$ represent the univariate background error covariances for the chemical
concentrations and emissions, respectively, while $\mathbf{B}_{ec}$ is their cross-covariance.

By decomposing $\mathbf{B}_{cc}$ as $\mathbf{B}_{cc} = \mathbf{B}_{cc}^{1/2} \mathbf{B}_{cc}^{\mathrm{T}/2}$, where $\mathbf{B}_{cc}^{1/2}$ is the square root of $\mathbf{B}_{cc}$, the control vector transform $\Delta \mathbf{c} = \mathbf{B}_{cc}^{1/2} \boldsymbol{\chi}$
can be defined, so that the cost function in Eq. (2) can be expressed in terms of the control vector $\boldsymbol{\chi}$ as

$$J(\boldsymbol{\chi}) = \frac{1}{2} \boldsymbol{\chi}^{\mathrm{T}} \boldsymbol{\chi} + \frac{1}{2} (\mathbf{d} - \mathbf{H} \mathbf{B}_{cc}^{1/2} \boldsymbol{\chi})^{\mathrm{T}} \mathbf{R}^{-1} (\mathbf{d} - \mathbf{H} \mathbf{B}_{cc}^{1/2} \boldsymbol{\chi}). \tag{A3}$$

Assuming that $\mathbf{B}_{ee}$ can similarly be decomposed as $\mathbf{B}_{ee} = \mathbf{B}_{ee}^{1/2} \mathbf{B}_{ee}^{\mathrm{T}/2}$ such that $\mathbf{B}_{cc}^{1/2}$ and $\mathbf{B}_{ee}^{1/2}$ share the same basis, then from
Eq. (3) the emissions increment can be found from the transformation $\Delta \mathbf{e} = \mathbf{B}_{ee}^{1/2} \boldsymbol{\chi}$. In summary, the emissions inversion
consists of two steps: First, the cost function in Eq. (A3) is minimized as a function of the control vector $\boldsymbol{\chi}$, followed by using
the value of $\boldsymbol{\chi}$ found in the minimization in the transform $\Delta \mathbf{e} = \mathbf{B}_{ee}^{1/2} \boldsymbol{\chi}$ to yield the emissions increment.

To construct background error covariances from an ensemble, we define the ensemble perturbation matrices $\delta \mathbf{C}$ and $\delta \mathbf{E}$ as

$$\delta \mathbf{C} = \frac{1}{\sqrt{K-1}} \begin{bmatrix} \delta \mathbf{c}_1 & \dots & \delta \mathbf{c}_K \end{bmatrix}, \tag{A4a}$$

$$\delta \mathbf{E} = \frac{1}{\sqrt{K-1}} \begin{bmatrix} \delta \mathbf{e}_1 & \dots & \delta \mathbf{e}_K \end{bmatrix}, \tag{A4b}$$

where the vectors $\delta \mathbf{c}_i$ and $\delta \mathbf{e}_i$ are the concentration and emissions perturbations for the $i^{\mathrm{th}}$ member of an ensemble of size $K$.
We take these perturbations as

$$\delta \mathbf{c}_i = \mathcal{M}(\mathbf{e}_i, \boldsymbol{\theta}) - \mathcal{M}(\mathbf{e}^{\mathrm{b}}, \boldsymbol{\theta}), \tag{A5a}$$



$$\delta\mathbf{e}_i = \mathbf{e}_i - \mathbf{e}^{\mathrm{b}}, \tag{A5b}$$

where $\mathbf{e}^{\mathrm{b}}$ are the background (i.e. original) emissions. This is the formulation of 4DEnVar (using the nomenclature of Bannister (2017)), but constructs the ensemble perturbations by subtracting the background instead of the ensemble mean. The localization matrix $\mathbf{L}$ can be written in block form as

$$\mathbf{L} = \begin{bmatrix} \mathbf{L}_{cc} & \mathbf{L}_{ce} \\ \mathbf{L}_{ec} & \mathbf{L}_{ee} \end{bmatrix}, \tag{A6}$$

where $\mathbf{L}_{ec} = \mathbf{L}_{ce}^{\mathrm{T}}$. The ensemble-based background error covariance is then constructed as

$$\mathbf{B} = \begin{bmatrix} \mathbf{L}_{cc} \circ (\delta\mathbf{C}\delta\mathbf{C}^{\mathrm{T}}) & \mathbf{L}_{ce} \circ (\delta\mathbf{C}\delta\mathbf{E}^{\mathrm{T}}) \\ \mathbf{L}_{ec} \circ (\delta\mathbf{E}\delta\mathbf{C}^{\mathrm{T}}) & \mathbf{L}_{ee} \circ (\delta\mathbf{E}\delta\mathbf{E}^{\mathrm{T}}) \end{bmatrix}, \tag{A7}$$

where $\circ$ is the Hadamard (element-wise) product. The square roots of the background error covariances, which are used in the cost function given in Eq. (A3) and the emissions control vector transform, can be expressed as (Bishop et al., 2011)

$$\mathbf{B}_{cc}^{1/2} = \frac{1}{\sqrt{K-1}} \left[ \mathrm{diag}(\delta\mathbf{c}_1)\mathbf{L}_{cc}^{1/2} \quad \dots \quad \mathrm{diag}(\delta\mathbf{c}_K)\mathbf{L}_{cc}^{1/2} \right], \tag{A8a}$$

$$\mathbf{B}_{ee}^{1/2} = \frac{1}{\sqrt{K-1}} \left[ \mathrm{diag}(\delta\mathbf{e}_1)\mathbf{L}_{ee}^{1/2} \quad \dots \quad \mathrm{diag}(\delta\mathbf{e}_K)\mathbf{L}_{ee}^{1/2} \right], \tag{A8b}$$

where $\mathbf{L}_{cc}$ and $\mathbf{L}_{ee}$ have been decomposed as $\mathbf{L}_{cc} = \mathbf{L}_{cc}^{1/2}\mathbf{L}_{cc}^{\mathrm{T}/2}$ and $\mathbf{L}_{ee} = \mathbf{L}_{ee}^{1/2}\mathbf{L}_{ee}^{\mathrm{T}/2}$, respectively, and $\mathrm{diag}(\cdot)$ is the diagonalization operator.

*Author contributions.* Michael Sitwell performed all emissions inversions, conducted the analysis, and wrote the manuscript. Mark Shephard ran all CrIS ammonia inversions used in this work. Both authors jointly performed the quality control for the CrIS ammonia retrievals and prepared the observation files for use in the inversion system.

*Competing interests.* The authors declare that they have no conflicts of interest.

*Acknowledgements.* The authors would like to thank Karen Cady-Pereira and Enrico Dammers for their role in the development of the CrIS ammonia retrieval algorithm and the processing of the retrievals used for this work. We also thank Yves Rochon and Mike Moran for providing feedback on drafts on this manuscript, Verica Savic-Jovcic for the set up of GEM-MACH used in this work, Alex Lupu for





providing help and expertise in utilizing surface observation, Qiong Zheng for providing critical-load fields, and the GEM-MACH team for

the development and use of the model. We thank Yves Rochon, Mike Moran, Paul Makar, Verica Savic-Jovcic, Alex Lupu, Junhua Zhang, Hansen Cao, Daven Henze, and Enrico Dammers for useful discussions that advanced the work done for this paper. We thank the NOAA Comprehensive Large Array-data Stewardship System (CLASS; Liu et al. (2014)), with special thanks to Axel Graumann (NOAA), for providing the CrIS Level 1 and Level 2 CrIS REDRO and NUCAPS input atmospheric state data. We thank ECCC for the use of daily filter pack and precipitation-chemistry observations from the CAPMoN network, as well as the provincial, territorial, and regional government

NAPS partners for the use of their ambient air quality data. We thank the NADP for the use of observations from the AMoN, NTN, and AIRMoN networks, the US EPA for use of data from the CASTNET and CSN networks, as well as observations made available through the AQS, the US EPA and National Park Service and other federal, state, and tribal partners for use of observations from the IMPROVE network.





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
