# Peer review of "An Ensemble-Variational Inversion System for the Estimation of Ammonia Emissions using CrIS Satellite Ammonia Retrievals"

_Atmospheric Chemistry and Physics, 2021_

## Author Comment (AC1)

[Figure]

[Figure]

**Figure 1.** Number of CrIS ammonia retrievals used in the inversions within $0.5° \times 0.5°$ longitude/latitude bins for May to August 2016. The total number of retrievals for each month is displayed in the lower right corner of each panel.

[Figure]

**Figure 2.** Comparisons of ammonia surface observations from the NAPS and AMoN networks with GEM-MACH surface fields. The left and center columns show bias values for each station when the original and updated ammonia emissions are used, respectively. The right column shows the relative improvement of the root-mean-square error (RMSE) for each station. NAPS and AMoN stations are denoted with circular and triangular markers, respectively.

[Figure]

**Figure 3.** Monthly mean vertical profiles near (a) Toronto, Ontario and (b) Grand Junction, Colorado for May 2016. Plots on the left show mean profiles for GEM-MACH run with the original ammonia emissions, while plots on the right show mean differences between the updated and original runs. The vertical coordinate $\sigma$ is the ratio of the pressure to the surface pressure. The legend in the upper right plot applies to all plots.

---

## Author Response (AR1)

Thank you for your comments. Reviewer comments are in italics.

1) *I am not sure why the author spent much effort about the difference between log-space and linear-space H(observational operator) to justify the 'hybrid' technique. Is that because of the scientific importance? If that is the efficient approach, then 3.3 should be shortened and briefly explain the benefit of the compromised approach. (Those descriptions and testing results are too technical to this journal)*

The hybrid technique is a novel comparison method that has a significant impact on the inversions, as shown in the 'Results' section. The discussion around this technique is to emphasize that the comparison method chosen can dramatically change the results of the inversions. As this is a new technique, we would like the details of this method documented, but have moved more of the details to a new appendix (Appendix B) to streamline the main text. Additionally, the plot showing the operator selection using the hybrid method (what was labeled as Fig. 3) has been moved to the Supplement.

In the interactive discussion, Anonymous Referee #2 requested additional details of the hybrid technique (under the comment 'Sensitivity of constants' in the comments RC1) and in response we actually added details of the hybrid method to a subsequent version of the manuscript. Moving more of the details to a new appendix and Fig.3 to the Supplement is our attempt at trying to balance these comments with the comments from Anonymous Referee #2.

2) *I understand why the column comparisons with the averaging kernels for this work. But if the operator has higher sensitivities with the vertical profile, is that any possibility to compare the satellite data and model at a specific level only with the highest sensitivity (such as 700hPa or near-surface levels only)?*

I would image that this approach possibly could yield fairly similar results to using the full profile (assuming the a priori was still accounted for). While many profiles are sharply peaked at a particular level, not all profiles have such a narrow peak. Also, different profiles will peak at different levels. So since we had access to the full profile, we used the information from the full profile.

3) *The author has to comment more about the reason for GEM-MACH performance before and after the inversion since readers do not know much about the potential weakness or biases of the generic model performance. We don't determine the meaning of changes by this work well.*

To help address this comment, in addition to adding more discussion on this point, we thought it would be beneficial to rearrange the 'Results' section to better highlight these differences. Previously, the 'Results' section was subdivided into subsections by result type (i.e. a subsection

looking at the inversion result, another subsection looking at the surface NH3, etc…). In the revised manuscript, the 'Results' section instead starts with a subsection describing the 'before' case (Subsection 4.1), followed by a subsection describing the inversion (Subsection 4.2), then by a subsection describing the 'after' case (Subsection 4.3). Additional comments on the results from specific ground stations, biases, and emissions sources were also added (see 'Tracked Changes' version of the manuscript).

4) *The ammonia has a relatively short lifetime and the author claimed that the ammonia concentrations have increased. How is the degree of underestimation of NH3 emissions and the trends over the other continents? The comparison of this work to other regions(or studies) will be informative as well.*

Currently there are limited ammonia inversion studies outside of North America.

One study that examines ammonia inversions over Europe is currently under review for 'Journal of Geophysical Research: Atmospheres' (https://doi.org/10.1002/essoar.10507960.1). While the authors of this paper are coauthors on the JGR manuscript as well, the JGR manuscript uses a different model (GEOS-Chem) and inversion method (4D-Var). In the case of the unidirectional flux scheme, the inversion increases emissions in most of Europe in the Spring and Summer (with areas like Northern Italy as an exception).On the other hand, emissions are decreased in many places in Europe during the fall and winter. However, the annual emissions are increased by the inversions in most of Europe.

Another relevant manuscript that is currently under review in ACP and available in preprint is 'Data assimilation of CrIS-NH3 satellite observations for improving spatiotemporal NH3 distributions in LOTOS-EUROS' (https://doi.org/10.5194/acp-2021-473). This study uses CrIS NH3 retrievals to estimate NH3 emissions over Germany and parts of Belgium, and the Netherlands. Assimilation done with a LETKF increased emissions throughout this region, with increases as much as 30% for the total emissions over 2014-2018.

Response to Anonymous Referee #2

Thank you for your comments. Reviewer comments are in italics.

Note: Below, references to figure, table, and line numbers are to the previous version of the manuscript.

Major comments:

1) Evaluation:

*Mean bias evaluation can be misleading without also evaluating absolute error (ME or RMSE) due to the possibility of positive and negative biases canceling each other*

This concern was already addressed in the paper as the standard deviation of differences were computed for these statistics, displayed in Tables S1-S5, and commented on lines 440-445, 553-556, and 664-665 of the paper. As discussed at the end of Section 2.2, the bias, standard deviation of differences, and correlation coefficients were computed for all data sets, all of which are displayed in Tables S1-S5. Any cancelling of errors will be reflected in the standard deviation of differences. Note that the RMSE can easily be computed by the reader by adding the NMB and NSTD in quadrature and then taking the square root. I had included the RMSE in Tables S1-S5 in an earlier draft of the paper, but decided to remove it because the tables were too big to fit in the page, which were already in landscape. Since the RMSE is redundant information if the bias and standard deviation are already given, I decided to remove the RMSE from the tables (although it would have been nice to display).

*Throughout the results, total NMB (e.g. Figures 7, 11, 13, 14,16) from all sites are used to demonstrate the impacts of the updated ammonia emissions using the inversion approach… RMSE should be presented in the paper with NMB figures (7, 11, 13, 14, 16).*

The reason more emphasis is given to the NMB as compared to NSTD (or RMSE, as well as the correlation coefficient) is that the changes in the NSTD were statistically insignificant for all cases examined, with only one exception (comparison with the log-space operator in June for AMoN). All differences between the NSTD of original and updated hybrid cases were statistically insignificant, which can be seen by looking at the 'sig' column of the NSTD in Tables S1-S5. This was discussed on lines 440-445, 553-556, and 664-665 of the paper. For this reason, I chose not to include the NSTD in Figures 7, 11, 13, 14, and 16, with the thought that the descriptions in the lines referenced above would be sufficient considering the results.

If RMSE is plotted, the differences between in the RMSE are a mixture between the statistically significant differences in the biases and the statistically insignificant differences in the NSTD. This is why NMB and NSTD were displayed separately. Although displaying the RMSE can be nice, as it yields a single number for comparison, it is redundant with the bias and NSTD taken together. As including plots of either the RMSE or NSTD would greatly increase the number of plots in the paper, given that the changes in NSTD were statistically insignificant and already described in the text, we chose not to include these plots.

With comments made on lines 440-445, 553-556, and 664-665 directing the reader to the NSTD results, it should be clear that taking the NMB and NSTD results together constitutes an evaluation of the absolute error and that the emphasis given to the NMB is simply due to the statistical significance of these results.

What was previously labeled as Fig. S3, which showed individual biases and RMSE values for each station, was moved to the main text of the manuscript to further highlight results for the

RMSE. More discussion of this point was also added in what now is labeled as Section 4.3 (see 'Tracked Changes' version of the manuscript).

2) Sensitivity of constants:

Before discussing this point, I'd like to address a misreading of Figs. S3, S5, S6, S7, and S8.

*For instance, based on Figure S3, sites in the western central U.S. (around Colorado) tend to have worse performance*

…

*Why do all RMSE (updated) / RMSE (original) ratio figures (S3, S5…, S8) in supplement have negative values?*

Note that the right columns in Figs. S3, S5, S6, S7, and S8 are labeled as '1 − RMSE(updated)/RMSE(original)'. I think the '1 - ' might have been missed. So the sites around Colorado show better performance, not worse. I also assume that in the comment made on evaluation that

*RMSE is presented in the supplement figures (Figures S3, S5, S6, S7, S8) and it seems that many sites show worse performance.*

that the same mistake has been made. I'm not sure how much correcting this misreading will change these comments, but I will try to respond to these comments the best I can given the situation. I have reformatted this title to try to make it more legible and fit better within the column.

*Sensitivity analysis on the constants used in the hybrid approach seems to be important and useful for the "ideal" constant selection.*

In this study, the results do not appear to be very sensitive to the chosen parameters for the hybrid method. I tried lowering the value of X_min by a factor of 10, which only changed about 0.02% of retrieval comparisons for May-August 2016. The locations of these retrievals were also reasonably spread out over the model domain, so this change is unlikely to have much of an influence in the inversion at any location. I tried lowering X_min by another factor of 10, which showed almost the same differences of 0.02%. I only tried lowering X_min since increasing it, say by a factor of 10, would start to label some non-negligible profiles as negligible, which is not desirable. When lowering c_min by a factor of 10 (keeping X_min at its original value), 0.7% of retrieval comparisons change, again spread out over the model domain. Although this change effects more profiles, it is still a small number of retrievals, and not localized in any particular location. I have added this text at the end of Section 3.3:

"For the time period and locations examined in this study, the hybrid comparison method does not appear to be particularly sensitive to the values chosen for $X_{\mathrm{min}}$ and $c_{\mathrm{min}}$ for values smaller than those chosen here. Reducing $X_{\mathrm{min}}$ and $c_{\mathrm{min}}$ by an order of magnitude only changes the operator selected for less than 1\% of retrieval/model pairs, which were spread out throughout the model domain. While reducing the values of $X_{\mathrm{min}}$ and $c_{\mathrm{min}}$ yielded little difference in the retrieval-to-model comparison, selecting significantly higher values for these parameters would result in classifying some non-negligible profiles as negligible, and so must be done with caution."

*Maybe the values used for linear-log cutoff should be variable spatially or temporally depending on the ground sources.*

The parameters used in the hybrid method are used to detect model profiles with non-negligible amounts of ammonia that have been 'zeroed out' by the log-space averaging kernel. As such, the method's parameters X_min and c_min, are used to define a minimum non-negligible profile. I'm not quite sure what the motivation would be to have these parameters vary in space or time given their physical interpretation. Having them varying in time or space would imply that what you consider to be the minimum non-negligible profile varies in space or time as well, and am not quite sure why this would be a desirable property. However, since the hybrid method does not seem to be very sensitive to the chosen values for X_min and c_min to begin with, this might be a moot point.

Minor comments:

1) *92 line: what does "The number of degrees of freedom for this retrieval is 0.956" mean?*

I'm assuming this question means "What does 'degree of freedom' of a retrieval mean"? The degrees of freedom for a signal is a very frequently used diagnostic quantity of a retrieval, which is the number of independent pieces of information that could be measured in the retrieval process.

2) *May to August 2016 study. Since this approach is developed for the GEM-MACH air quality forecasting model, probably it is important to evaluate how this approach performs in other seasons with cooler temperature and low ammonia emissions as well.*

We agree. For this initial study, demonstrating the proof-of-concept for the NH3 inversion method, as well as the model-to-retrieval comparison method, we focused on the warmer months across North America as these conditions are more favourable for infrared satellite ammonia retrievals (higher concentrations of ammonia and greater thermal contrast

between the surface and the atmosphere). More evaluations are planned for the future that will cover the whole year, including the cooler seasons with less ammonia emissions.

3) *In reality, fires exist, and fire emissions are included in forecasting. Is this approach appropriate if weekly updates are applied for emissions under fire conditions?*

How fires are handled depends on context. For instance, if the inversions are going to be used to update emissions to be used for a different year, then if a fire significantly impacts the inversion, then fires from one year may effect the prescribed emissions in a different year, which may not be desirable. If instead the inversions are only being used for the same time period, then having the fire emissions significantly influence the inversion could be desirable. I have added these lines in Section 2.1:

"… due to forest fires with other emission sources. While we seek to minimize the effect of forest fires on the emissions inversions in this work, in other contexts this might not be necessary or desirable. For example, if the emissions are only used for the time period when the fire occurred, having the fires affect the inversion may be advantageous."

4) *What is magnitude of the issue related to the non-detection of ammonia discovered on the quality of CrIS data which affects non-source regions in the domain?*

As the focus of this study is on source regions, this issue related to non-detects does not have a significant effect on the CrIS retrievals used in this study. However, at high northern latitudes far away from significant sources, this becomes an important issue. So the impact of this on the inversions performed for this work is small, but could be an important issue if instead we focused on remote non-source regions. As the newer version of the CrIS $NH_3$ retrievals included non-detects, this allows for the possibility of focussing more on remote non-source regions for future work.

5) *70–85% of the retrievals used in the inversions coming from daytime retrievals. What causes nighttime retrievals to have low quality?*

During the development of the version of the CrIS $NH_3$ retrieval product used, it was found that performing retrievals over areas with temperature inversions near the surface was challenging. This could be in part due to not having adequate a priori profiles for these situations. For this reason, a quality flag was added to filter these retrievals. As these situations occur more frequently during the night, this quality flag removed a large fraction of the nighttime retrievals, while removing a much smaller fraction of the daytime retrievals. Also, as the $NH_3$ signal is generally higher during the daytime, as non-detects were not included in this product version, more of the retained retrievals were for the daytime.

6) *Figure S1: hard to tell difference among 0 to 50 color scale in plots.*
   I have rescaled the colour bar from 0 to 100 so that the 0 to 50 portion is easier to read. I tried out different colour maps to see if it made it easier to read, but didn't find that they improved the readability much. I also tried a log scale, but it then made the higher end of the colour scale harder to read.

7) *Figure 9a – NH3 value higher than graph horizontal range.*

   The x-axis range has been extended. New figure is attached.

8) *Why do all RMSE (updated) / RMSE (original) ratio figures (S3, S5…, S8) in supplement have negative values?*

   Response to this comment is in (2) of the major comments section.

9) *The inclusion of the critical load exceedances seems to be out of the focus for this paper although updating ammonia emissions affects N deposition. Given the purpose of the paper and more thorough evaluation needed for this new approach, it is recommended to remove the critical load results from the paper*

   All references to the critical load have been removed from the manuscript.

---

## Author Response (AR2)

Reviewer comments are in italics.

*1. Line 83 – "using version 1.5 of the retrieval algorithm", reference on this version?*

Citation to Shephard et al. 2020 added here.

*2. Line 218 – "apply it an emissions inversion" to "apply it to an emissions inversion"*

Changed.

*3. Line 292 – "temporal profiles as described in Section 2.3": do you mean that you used the temporal profiles from SMOKE to allocate the monthly disturbed emissions for GEM. For an ensemble of 100 members (Line 242) , I assume that you run SMOKE and GEM-MACH for 100 times at ~10km resolution, right? I am wondering whether resources and time required for the 100 times SMOKE and GEM-MATCH simulations impose hurdles in the operational air quality modeling or forecast, particularly when you try to shorten the time lag and reduce inversion time window from a month to 1-2 weeks as mentioned in the conclusion.*

Here we mean that the intra-month temporal distribution of the emissions are the same for all emissions sets. These temporal profiles are set via the input profiles to SMOKE and the processing done by SMOKE, but since the same profiles are used for every emissions set, SMOKE only has to be run once. In practice, after the original set of emissions was created by SMOKE, temporal profiles for each grid point for each month were computed. When a new monthly mean emission set was computed, these pre-computed profiles were used to create the hourly emission files. The application of the pre-computed profiles takes relatively little computing resources.

*4. Line 359 – "In summary, the CrIS retrieval is compared to the GEM-MACH model by computing the difference between the total column". For ground level emission perturbation, it affects the surface (or ground) level (highest pressure level) NH3 concentration most (Lines 380 – 381) . I would think comparing surface level GEM-MACH to the surface level CrIS makes more sense. Thus, you can avoid upper level concentrations resulting from transport. Do you have any comments on the reason for choosing the whole column NH3 for comparison instead of the surface NH3?*

Comparing only the surface level concentrations would likely give reasonable results. However, although many retrievals have their highest sensitivity at the surface, this is not true for all retrievals. Some retrievals have averaging kernels that peak much higher above the surface (most peak by 700 hPa, but a few peak even higher above the surface). So using these retrievals in an inversion that only compares the surface values would not be optimal. One could simply not use these retrievals in the inversion, but we wanted to maximize the information used in the inversions. For retrievals that do peak at or near the surface, the averaging kernels dampen the influence of levels farther above the surface in the inversion, so I wouldn't expect much of a difference in effect that these retrievals have on the inversion if only the surface was used instead.

*5. For the Fig 6 plots on the right, I am suggesting that % of sites with positive value (percent of improved*

*sites – reduced RMSE) is placed on the lower right corner of each plot (like Fig 4 plots) as it is hard to tell overall performance based on 1- RMSE/RMSE plots now.*

Done.

*6. Lines 397-398 – "displayed in Figure 7, which shows the monthly mean column values within 0.5. ×0.5 longitude/latitude bins …" Does this mean that the ammonia emissions inversions are conducted at 0.5X0.5 long/lat bin resolution? If yes, I do not recall this information is stated explicitly in the paper. Does the 0.5 bin selected have something to do with 40km mentioned in Line 297? The GEM-MACH simulation is at ~10km resolution and CrIS resolution is ~14k at nadir. If inversion is conducted at 0.5 bin, what does it mean to emissions of cells within 0.5 bin for GEM-MACH?*

The 0.5 x 0.5 longitude/latitude bins were used here for display purposes only and not used anywhere in the inversion. Looking back at this plot, I can see that the choice of this binning for this plot may have been confusing. I have changed this figure so that the data is binned using the GEM-MACH horizontal grid instead. There were a few other plots that used the 0.5 x 0.5 longitude/latitude bins, which I have changed as well to reduce the potential of confusion.

*7. Results from 4.1, 4.2, 4.3, 4.4 subsections seem to be all mixed together, such as that figures in 4.1 show results for 4.3. Here are the suggested sub-sections with deposition at the end instead of being between the two PM subsections:*

*4.1 -- Emissions Inversions: discuss results for fig 7 and 3. I don't think 4.4 should be in the result section because it is related to your emission inversion approach development described in Section 3. If you want to demonstrate the impacts of the selections on emissions, it is probably better to have it in this sub-section.*

*4.2 - Effect of Inversions on Model Ammonia: Combine your original 4.1 Model Ammonia Performance Without Inversion into this subsection to reduce redundance. In analyzing the effect of the inversion, the model results with/without the inversions are compared and evaluated.*

*4.3 - Impacts on PM Formation*

*4.4 - Impact on PM Size Distribution*

*4.5 - Impact on Deposition*

The results section was reorganized to a structure similar to that suggested.